# Understanding Model Ensemble in Transferable Adversarial Attack

**Wei Yao** [* 1 2 3]   **Zeliang Zhang** [* 4]   **Huayi Tang** [1 2 3]   **Yong Liu** [1 2 3]

## Abstract

Model ensemble adversarial attack has become a powerful method for generating transferable adversarial examples that can target even unknown models, but its theoretical foundation remains underexplored. To address this gap, we provide early theoretical insights that serve as a roadmap for advancing model ensemble adversarial attack. We first define transferability error to measure the error in adversarial transferability, alongside concepts of diversity and empirical model ensemble Rademacher complexity. We then decompose the transferability error into vulnerability, diversity, and a constant, which rigidly explains the origin of transferability error in model ensemble attack: the vulnerability of an adversarial example to ensemble components, and the diversity of ensemble components. Furthermore, we apply the latest mathematical tools in information theory to bound the transferability error using complexity and generalization terms, validating three practical guidelines for reducing transferability error: (1) incorporating more surrogate models, (2) increasing their diversity, and (3) reducing their complexity in cases of overfitting. Finally, extensive experiments with 54 models validate our theoretical framework, representing a significant step forward in understanding transferable model ensemble adversarial attacks.

---

[*]Equal contribution [1]Gaoling School of Artificial Intelligence, Renmin University of China, Beijing, China [2]Beijing Key Laboratory of Research on Large Models and Intelligent Governance [3]Engineering Research Center of Next-Generation Intelligent Search and Recommendation, MOE [4]Independent Researcher. Contributed ideas during the author's B.S. studies at Huazhong University of Science and Technology, Wuhan, China. Correspondence to: Yong Liu <liuyonggsai@ruc.edu.cn>.

*Proceedings of the 42$^{nd}$ International Conference on Machine Learning*, Vancouver, Canada. PMLR 267, 2025. Copyright 2025 by the author(s).

## 1 Introduction

Neural networks are highly vulnerable to adversarial examples (Szegedy et al., 2013; Goodfellow et al., 2014)—perturbations that closely resemble the original data but can severely compromise safety-critical applications (Zhang & Li, 2019; Kong et al., 2020; Bortsova et al., 2021). Even more concerning is the phenomenon of adversarial transferability (Papernot et al., 2016; Liu et al., 2017): adversarial examples crafted to deceive one model often succeed in attacking others. This property enables attacks without requiring any knowledge of the target model, significantly complicating efforts to ensure the robustness of neural networks (Dong et al., 2019; Silva & Najafirad, 2020).

To enhance adversarial transferability, researchers have proposed a range of algorithms that fall into three main categories: input transformation (Xie et al., 2019; Wang et al., 2021), gradient-based optimization (Gao et al., 2020; Xiong et al., 2022), and model ensemble attacks (Li et al., 2020; Chen et al., 2024b). Among these, model ensemble attacks have proven especially powerful, as they leverage multiple models to simultaneously generate adversarial examples that exploit the strengths of each individual model (Dong et al., 2018). Moreover, these attacks can be combined with input transformation and gradient-based optimization methods to further improve their effectiveness (Tang et al., 2024). However, despite the success of such attacks, their theoretical foundation remains poorly understood. This prompts an important question: *Can we establish a theoretical framework for transferable model ensemble adversarial attacks to shape the evolution of future algorithms?*

To conduct a preliminary exploration of this profound question, we propose three novel definitions as a prerequisite of our theoretical framework. Firstly, we define *transferability error* as the gap in expected loss between an adversarial example and the one with the highest loss within a feasible region of the input space. It captures the ability of an adversarial example to generalize across unseen models, representing its transferability. Secondly, we introduce *prediction variance* across the ensemble classifiers. It offers a novel perspective on quantifying diversity in model ensemble attacks, providing a fresh approach to guide the selection of ensemble components. Finally, we also introduce the *empirical model ensemble Rademacher complexity*, inspired

by Rademacher complexity (Bartlett & Mendelson, 2002), as a measure of the flexibility of ensemble components.

With these three definitions, we offer two key theoretical insights. First, we show the ***vulnerability-diversity decomposition*** of transferability error (Figure 1), highlighting the preference for ensemble components that are powerful attackers and induce greater prediction variance among themselves. However, this also uncovers a *fundamental trade-off between vulnerability and diversity*, making it challenging to maximize both simultaneously. To mitigate this issue and provide more practical guidelines, we present an upper bound for transferability error, incorporating empirical model ensemble Rademacher complexity and a generalization term. The primary challenge in proof lies in the application of cutting-edge mathematical tools from information theory (Esposito & Mondelli, 2024), which are crucial for addressing the complex issue of relaxing the independence assumption among surrogate classifiers. Our theoretical analysis leads to a crucial takeaway for practitioners: Including ***more and diverse surrogate models*** with ***reduced model complexity in cases of overfitting*** helps tighten the transferability error bound, thereby improving adversarial transferability. Finally, the experimental results support the soundness of our theoretical framework, highlighting a key step forward in the deeper understanding of transferable model ensemble adversarial attacks.

## 2 Related Work

### 2.1 Transferable Adversarial Attack

Researchers have developed various algorithms to enhance adversial transferability. Most of them fall into three categories: input transformation, gradient-based optimization, and model ensemble attack. **Input transformation** techniques apply data augmentation strategies to prevent overfitting to the surrogate model. For instance, random resizing and padding (Xie et al., 2019), downscaling (Lin et al., 2019), and mixing (Wang et al., 2021). **Gradient-based optimization** optimizes the generation of adversarial examples to achieve better transferability. Some popular ideas include applying momentum (Dong et al., 2018), Nesterov accelerated gradient (Lin et al., 2019), scheduled step size (Gao et al., 2020) and gradient variance reduction (Xiong et al., 2022). **Model ensemble attack** combine outputs from surrogate models to create an ensemble loss, increasing the likelihood to deceive various models simultaneously. It can be applied collectively with both input transformation and gradient-based optimization algorithms (Tang et al., 2024). Some popular ensemble paradigms include loss-based ensemble (Dong et al., 2018), prediction-based (Liu et al., 2017), logit-based ensemble (Dong et al., 2018), and longitudinal strategy (Li et al., 2020). Moreover, advanced ensemble algorithms have been created to ensure better ad-

versarial transferability (Li et al., 2023; Wu et al., 2024; Chen et al., 2024b). An extended and detailed summary of related work is in Appendix A.

Within the extensive body of research on model ensemble attacks, two notable and intriguing observations stand out. First, increasing the number of models in an ensemble improves adversarial transferability (Liu et al., 2017; Dong et al., 2018; Lin et al., 2019; Gubri et al., 2022b; Liu et al., 2024). Second, using more diverse surrogate models with varying architectures and back-propagated gradients (Tang et al., 2024) further enhances transferability. However, to our best knowledge, these intriguing phenomena have yet to be fully understood from a theoretical perspective. In this paper, we present the first theoretical framework to explain these phenomena, providing actionable insights that pave the way for future algorithm design.

### 2.2 Theoretical Understanding of Adversarial Transferability

In contrast to the wealth of empirical and intuitive studies, research on the theoretical understanding of adversarial transferability remains limited. Recent efforts have primarily focused on aspects such as data (Tramèr et al., 2017), surrogate model (Wang & Farnia, 2023), optimization (Yang et al., 2021; Zhang et al., 2024a; Chen et al., 2024b; Fan et al., 2024) and target model (Zhao et al., 2023). Tramèr et al. (2017) investigates the space of transferable adversarial examples and establishes conditions on the data distribution that suggest transferability for some basic models. In terms of the surrogate model generalization, Wang & Farnia (2023) builds the generalization gap to show that a surrogate model with a smaller generalization error leads to more transferable adversarial examples. From an optimization perspective, Yang et al. (2021); Zhang et al. (2024a) establish upper and lower bounds on adversarial transferability, linking it to model smoothness and gradient similarity. They suggest that increased surrogate model smoothness and less loss gradient similarity improve transferability. Chen et al. (2024b) provide theoretical evidence connecting transferability to loss landscape flatness and closeness to local optima. Fan et al. (2024) decompose adversarial transferability into local effectiveness and transfer-related loss, suggesting that flatness alone is insufficient to determine the whole picture of adversarial transferability. Regarding the target model, Zhao et al. (2023) theoretically reveal that reducing the discrepancy between the surrogate and target models can limit adversarial transferability.

Despite these theoretical advances, to the best of our knowledge, *transferable model ensemble adversarial attacks remain unexplored*. To address this gap, we take a pioneering step by presenting the first theoretical analysis of such attacks. Our work not only offers theoretical insights into

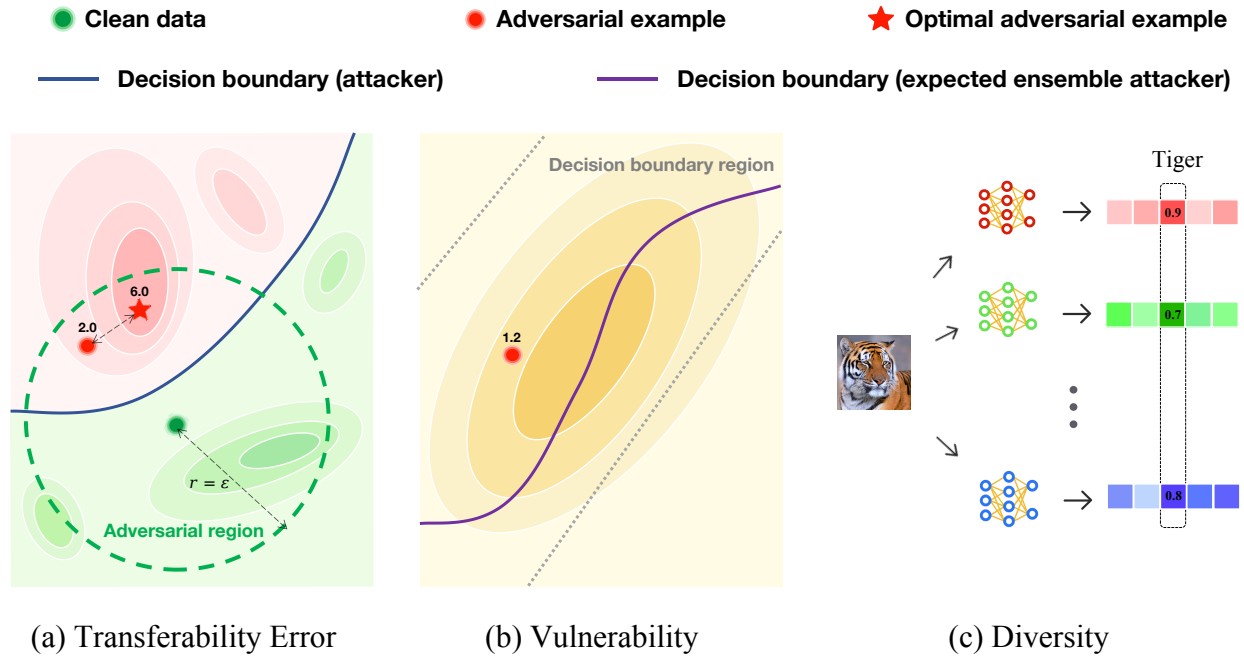

*Figure 1.* Vulnerability-diversity decomposition of transferability error. (a) The transferability error is defined as the difference in expected loss value between a given adversarial example and the most transferable one. (b) Vulnerability is the loss value of the expected ensemble classifier on the adversarial example. (c) Diversity is the variance in model ensemble predictions that correspond to the correct class.

these attacks but also incorporates recent advancements in information theory, laying the groundwork for future theoretical investigations into adversarial transferability.

## 3 Key Definitions: Transferability Error, Diversity, and Ensemble Complexity

In this section, we first highlight the fundamental goal of model ensemble adversarial attack (Section 3.1). Then we define the transferability error (Section 3.2), diversity in transferable model ensemble attack (Section 3.3) and empirical model ensemble Rademacher complexity (Section 3.4).

### 3.1 Model Ensemble Adversarial Attack

Given the input space $\mathcal{X} \subset \mathbb{R}^d$ and the output space $\mathcal{Y} \subset \mathbb{R}$, we have a joint distribution $\mathcal{P}_{\mathcal{Z}}$ over the input space $\mathcal{Z} = \mathcal{X} \times \mathcal{Y}$. The training set $Z_{\text{train}} = \{z_i | z_i = (x_i, y_i) \in \mathcal{Z}, y_i \in \{-1, 1\}, i = 1, \cdots, K\}$, which consists of $K$ examples drawn independently from $\mathcal{P}_{\mathcal{Z}}$. We denote the hypothesis space by $\mathcal{H} : \mathcal{X} \mapsto \mathcal{Y}$ and the parameter space by $\Theta$. Let $f(\theta; \cdot) \in \mathcal{H}$ be a classifier parameterized by $\theta \in \Theta$, trained for a classification task using a loss function $\ell : \mathcal{Y} \times \mathcal{Y} \mapsto \mathbb{R}_0^+$. Let $\mathcal{P}_\Theta$ represent the distribution over the parameter space $\Theta$. Define $\mathcal{P}_{\Theta^N}$ as the joint distribution over the product space $\Theta^N$, which denotes the space of $N$ such sets of parameters. We use $Z_{\text{train}}$ to train $N$ surrogate models $f(\theta_1; \cdot), \cdots, f(\theta_N; \cdot)$ for model ensemble. The training process of these $N$ classifiers can be viewed as sampling

the parameter sets $(\theta_1, \ldots, \theta_N)$ from the distribution $\mathcal{P}_{\Theta^N}$. For a clean data $\hat{z} = (\hat{x}, y) \in \mathcal{Z}$, an adversarial example $z = (x, y) \in \mathcal{Z}$, and $N$ classifiers for model ensemble attack, define the population risk $L_P(z)$ and the empirical risk $L_E(z)$ of the adversarial example $z$ as

$$L_P(z) = \mathbb{E}_{\theta \sim \mathcal{P}_\Theta}[\ell(f(\theta; x), y)], \qquad (1)$$

$$\text{and} \quad L_E(z) = \frac{1}{N} \sum_{i=1}^{N} \ell(f(\theta_i; x), y). \qquad (2)$$

Intuitively, a transferable adversarial example leads to a large $L_P(z)$ because it can attack many classifiers with parameter $\theta \in \Theta$. Therefore, the most transferable adversarial example $z^* = (x^*, y)$ around $z$ is defined as

$$x^* = \arg \max_{x \in \mathcal{B}_\epsilon(\hat{x})} L_P(z), \qquad (3)$$

where $\mathcal{B}_\epsilon(\hat{x}) = \{x : \|x - \hat{x}\|_2 \leq \epsilon\}$ is an adversarial region centered at $\hat{x}$ with radius $\epsilon > 0$. However, the expectation in $L_P(z)$ cannot be computed directly. Thus, when generating adversarial examples, the empirical version Eq. (2) is used in practice, such as loss-based ensemble attack (Dong et al., 2018). Therefore, the adversarial example $z = (x, y)$ is obtained from the following equation

$$x = \arg \max_{x \in \mathcal{B}_\epsilon(x)} L_E(z). \qquad (4)$$

There is a gap between the adversarial example $z$ we find and the most transferable one $z^*$. It is due to the fact that the

ensemble classifiers cannot cover the whole parameter space of the classifier, i.e., there is a difference between $L_P(z)$ and $L_E(z)$. Accordingly, the core objective of transferable model ensemble attack is to design approaches that approximate $L_E(z)$ to $L_P(z)$, thereby increasing the transferability of adversarial examples.

## 3.2 Transferability Error

Considering the difference between $z$ and $z^*$, the transferability of $z$ can be characterized as the difference in population risk between it and the optimal one.

**Definition 3.1** (Transferability Error). The transferability error of $z$ with radius $\epsilon$ is defined as:

$$TE(z, \epsilon) = L_P(z^*) - L_P(z). \tag{5}$$

There always holds $TE(z, \epsilon) \geq 0$ as $L_P(z^*) \geq L_P(z)$. The closer $TE(z, \epsilon)$ is to 0, the better the transferability of $z$. Therefore, in principle, the essential goal of various model ensemble attack algorithms is to make transferability error $TE(z, \epsilon)$ as small as possible. Moreover, if the distribution over the parameter space $\mathcal{P}_\Theta$, adversarial region $\mathcal{B}_\epsilon(x)$ and loss function $\ell$ are fixed, then $L_P(z^*)$ becomes a constant, which means that the goal of minimizing $TE(z, \epsilon)$ becomes maximizing $L_P(z)$.

In the following lemma, we will show how the difference between empirical risk and population risk affects the transferability error of $z$. The proof is in Appendix C.1.

**Lemma 3.2.** *The transferability error defined by Eq. (5) is bounded by the largest absolute difference between $L_P(z)$ and $L_E(z)$, i.e.,*

$$TE(z, \epsilon) \leq 2 \sup_{z \in \mathcal{Z}} |L_P(z) - L_E(z)|. \tag{6}$$

The lemma strictly states that if we can bound the difference between $L_P(z)$ and $L_E(z)$, the transferability error can be constrained to a small value, thereby enhancing adversarial transferability. This indicates that we can develop strategies to make $L_E(z)$ closely approximate $L_P(z)$, ultimately improving the transferability of adversarial examples.

## 3.3 Quantifying Diversity in Model Ensemble Attack

Before the advent of model ensemble attacks, the formal definition of diversity in ensemble learning had remained a long-standing challenge for decades (Wood et al., 2024). While diverse intuitive definitions of diversity exist in the model ensemble attack literature (Li et al., 2020; Yang et al., 2021; Tang et al., 2024), we bridge the gap between transferable model ensemble attacks and recent advancements in ensemble learning theory (Ortega et al., 2022; Wood et al., 2024). Specifically, we propose measuring diversity among ensemble attack classifiers through prediction variance.

**Definition 3.3** (Diversity of Model Ensemble Attack). The diversity of model ensemble attack across $\theta \sim \mathcal{P}_\Theta$ for a specific adversarial example $z = (x, y)$ is defined as the variance of model prediction:

$$Var_{\theta \sim \mathcal{P}_\Theta} (f(\theta; x)) = \mathbb{E}_{\theta \sim \mathcal{P}_\Theta} [f(\theta; x) - \mathbb{E}_{\theta \sim \mathcal{P}_\Theta} f(\theta; x)]^2. \tag{7}$$

It indicates the degree of dispersion in the predictions of different ensemble classifiers for the same adversarial example. The diversity of model ensemble attack is a measure of ensemble member disagreement, independent of the label. From an intuitive perspective, the disagreement among the ensemble components helps prevent the adversarial example from overfitting to the classifiers in the ensemble, thereby enhancing adversarial transferability to some extent.

To calculate the diversity explicitly as a metric, we consider a dataset of adversarial examples $Z_{\text{attack}} = \{z_i | z_i = (x_i, y_i), i = 1, \cdots, M\}$ and $N$ classifiers in the ensemble. The diversity is computed as the average sample variance of predictions for all adversarial examples in the dataset:

$$\frac{1}{M} \sum_{i=1}^{M} \left[ \frac{1}{N} \sum_{j=1}^{N} \left( f(\theta_j; x_i) - \frac{1}{N} \sum_{j=1}^{N} f(\theta_j; x_i) \right)^2 \right].$$

*Remark.* For multi-class classification problems, $f(\theta; x)$ is replaced with the logit corresponding to the correct class prediction made by the classifier.

## 3.4 Empirical Model Ensemble Rademacher Complexity

We define the empirical Rademacher complexity for model ensemble by analogy to the original empirical Rademacher complexity (Koltchinskii & Panchenko, 2000; Bartlett & Mendelson, 2002).

**Definition 3.4** (Empirical Model Ensemble Rademacher Complexity). Given the input space $\mathcal{Z} = \mathcal{X} \times \mathcal{Y}$ and $N$ classifiers $f(\theta_1; \cdot), \cdots, f(\theta_N; \cdot)$. Let $\boldsymbol{\sigma} = \{\sigma_i\}_{i \in [N]}$ be a collection of independent Rademacher variables, which are random variables taking values uniformly in $\{+1, -1\}$. We define the empirical model ensemble Rademacher complexity $\mathcal{R}_N(\mathcal{Z})$ as follows:

$$\mathcal{R}_N(\mathcal{Z}) = \mathbb{E}_{\boldsymbol{\sigma}} \left[ \sup_{z \in \mathcal{Z}} \frac{1}{N} \sum_{i=1}^{N} \sigma_i \ell(f(\theta_i; x), y) \right]. \tag{8}$$

In conventional settings of machine learning, the empirical Rademacher complexity captures how well models from a function class can fit a dataset with random noisy labels (Shalev-Shwartz & Ben-David, 2014). A sufficiently complex function class includes functions that can effectively fit arbitrary label assignments, thereby maximizing

the complexity term (Mohri et al., 2018). Likewise, in model ensemble attack, Eq. (8) is expected to measure the complexity of the input space $\mathcal{Z}$ relative to the $N$ classifiers. Some extreme cases are analyzed in Appendix E.1.

# 4 Theoretically Reduce Transferability Error

## 4.1 Vulnerability-diversity Decomposition of Transferability Error

Inspired by the bias-variance decomposition (Geman et al., 1992; Domingos, 2000) in learning theory, we provide the corresponding theoretical support for prediction variance by decomposing the transferability error into vulnerability, diversity and constants. The proof and the empirical version of it is in Appendix C.2.

**Theorem 4.1** (Vulnerability-diversity Decomposition). *For a data point* $z = (x, y)$*, we consider the squared error loss* $l(f(\theta; x), y) = [f(\theta; x) - y]^2$*. Let* $\tilde{f}(\theta; x) = \mathbb{E}_{\theta \sim \mathcal{P}_\Theta} f(\theta; x)$ *be the expectation of prediction over the distribution on the parameter space. Then there holds*

$$TE(z, \epsilon) = L_P(z^*) - \underbrace{l(\tilde{f}(\theta; x), y)}_{Vulnerability} - \underbrace{Var_{\theta \sim \mathcal{P}_\Theta} f(\theta; x)}_{Diversity}.$$

$$(9)$$

*Remark.* A similar formulation also applies to the KL divergence loss in the multi-class classification setting, which is proved in Appendix C.3.

The "Vulnerability" term measures the risk of a data point $z$ being compromised by the model ensemble. If the model ensemble is sufficiently strong to fit the direction opposite to the target label, the resulting high loss theoretically reduces the transferability error. This insight suggests that *selecting strong attackers* as ensemble components leads to lower transferability error. The "Diversity" term implies that *selecting diverse attackers* in a model ensemble attack theoretically contributing to a reduction in transferability error. In conclusion, Theorem 4.1 provides the following guideline for reducing transferability error in model ensemble attack: we are supposed to choose ensemble components that are both strong and diverse. Theorem 4.1 connects the existing body of work and clarifies how each algorithm strengthens adversarial transferability. For instance, some approaches tend to optimizing the attack process (Xiong et al., 2022; Chen et al., 2023) to improve "Vulnerability", while others aim to diversify surrogate models (Li et al., 2020; 2023; Wang et al., 2024) to enhance "Diversity". Also, there are other definitions of diversity based on gradient in previous literature (Yang et al., 2021; Kariyappa & Qureshi, 2019). A more detailed discussion is presented in Appendix E.2.

However, due to the mathematical nature of Eq. (9), there remains a *vulnerability-diversity trade-off* in model ensem-ble attacks, similar to the well-known bias-variance trade-off (Geman et al., 1992). This means that, in practice, it is not feasible to maximize both "Vulnerability" and "Diversity" simultaneously. Recognizing this limitation, we proceed with further theoretical analysis to propose more guidelines for practitioners in the following section.

## 4.2 Upper Bound of Transferability Error

We develop an upper bound of transferability error in this section. We begin by taking Multi-Layer Perceptron (MLP) as an example of deep neural network and derive the upper bound of $\mathcal{R}_N(\mathcal{Z})$. The proof is in Appendix B.4.

**Lemma 4.2** (Ensemble Complexity of MLP). *Let* $\mathcal{H} = \{x \mapsto W_l \phi_{l-1} (W_{l-1} \phi_{l-2} (\ldots \phi_1 (W_1 x)))\}$ *be the class of real-valued networks of depth* $l$*, where* $x \in \mathbb{R}^{d_1}$*,* $W_i \in \mathbb{R}^{d_{i+1} \times d_i}$*. Given* $N$ *classifiers from* $\mathcal{H}$*, where the parameter matrix is* $W_{ij}, i \in \{1, \cdots, n\}, j \in \{1, \cdots, l\}$ *and* $T = \prod_{j=1}^l \sup_{i \in [n]} \|W_{i,j}\|_F$*. Let* $\|x\|_F \leq B$*. With 1-Lipschitz activation functions* $\phi_1, \cdots, \phi_{l-1}$ *and 1-Lipschitz loss function* $\ell(y f(x))$*, there holds:*

$$\mathcal{R}_N(\mathcal{Z}) \leq \frac{\left(\sqrt{(2 \log 2)l} + 1\right) BT}{\sqrt{N}}. \quad (10)$$

*Remark.* We also derive the upper bound of $\mathcal{R}_N(\mathcal{Z})$ for the cases of linear model (Appendix B.2) and two-layer neural network (Appendix B.3). These results are special cases of the above theorem.

In particular, a larger $N$ and smaller $T$ will give $\mathcal{R}_N(\mathcal{Z})$ a tighter bound. Notice that $T$ contains the norm of weight matrices, which is related to model complexity (Bartlett et al., 2017; Neyshabur et al., 2018). And a smaller model complexity corresponds to a smaller $T$ (Loshchilov & Hutter, 2019). In summary, Lemma 4.2 mathematically shows that *increasing the number of surrogate models* and *reducing the model complexity* of them can limit $\mathcal{R}_N(\mathcal{Z})$.

We now provide the upper bound of transferability error, and the proof is in Appendix C.4.

**Theorem 4.3** (Upper bound of Transferability Error). *Given the transferability error defined by Eq. (5) and general rademacher complexity defined by Eq. (8). Let* $\mathcal{P}_{\otimes_{i=1}^N \Theta}$ *be the joint measure induced by the product of the marginals. If the loss function* $\ell$ *is bounded by* $\beta \in R_+$ *and* $\mathcal{P}_{\Theta^N}$ *is absolutely continuous with respect to* $\mathcal{P}_{\otimes_{i=1}^N \Theta}$ *for any function* $f_i$*, then for* $\alpha > 1$ *and* $\gamma = \frac{\alpha}{\alpha - 1}$*, with probability at least* $1 - \delta$*, there holds*

$$TE(z, \epsilon) \leq 4\mathcal{R}_N(\mathcal{Z}) +$$

$$\sqrt{\frac{18\gamma\beta^2}{N} \ln \frac{2^{2 + \frac{1}{\gamma}} H_\alpha^{\frac{1}{\alpha}} \left(\mathcal{P}_{\Theta^N} \| \mathcal{P}_{\otimes_{i=1}^N \Theta}\right)}{\delta}}, \quad (11)$$

*where $H_\alpha(\cdot\|\cdot)$ is the Hellinger integrals (Hellinger, 1909) with parameter $\alpha$, which measures the divergence between two probability distributions if $\alpha > 1$ (Liese & Vajda, 2006).*

*Remark 1.* Our proposed setting where both the surrogate model and the target model adopt the same parameter space aligns with many realistic scenarios, as demonstrated in (Wu et al., 2024; Tang et al., 2024; Li et al., 2023; Xiong et al., 2022; Lin et al., 2019). However, Theorem 4.3 can be also extended to scenarios where the parameter distributions of surrogate model and target model differ. It is discussed in Appendix C.5 via a redefinition of the model space.

*Remark 2.* We provide further explanation of the Hellinger integral term $H_\alpha(\mathcal{P}_{\Theta^N}\|\mathcal{P}_{\otimes_{i=1}^N \Theta})$ in Appendix C.6.

*Remark 3.* Theorem 4.3 is grounded in the empirical model ensemble Rademacher complexity defined in Eq. (8). However, it can be extended to information-theoretic analysis with similar conclusions, as demonstrated in Appendix C.7.

The first term in Eq. (11) suggests that *incorporating **more surrogate models** with **less model complexity*** in ensemble attack will constrain $\mathcal{R}_N(\mathcal{Z})$ and enhances adversarial transferability. Intuitively, incorporating more models helps prevent any single model from overfitting to a specific adversarial example. Such theoretical heuristic is also supported by experimental results (Liu et al., 2017; Dong et al., 2018; Lin et al., 2019; Li et al., 2020; Gubri et al., 2022b; Chen et al., 2023; Liu et al., 2024), which also stress the advantage of more surrogate models to obtain transferable attack. Additionally, when there is an overfitting issue, models with reduced complexity will mitigate it.

The second term also suggests that a large $N$ (using more models) can lead to a tighter bound. Furthermore, it motivates the idea that reducing the interdependence among the parameters in ensemble components (i.e., increasing their diversity) results in a tighter upper bound for $TE(z, \epsilon)$. Recall that $H_\alpha(\mathcal{P}_{\Theta^N}\|\mathcal{P}_{\otimes_{i=1}^N \Theta})$ represents the divergence between the joint distribution $\mathcal{P}_{\Theta^N}$ and the product of marginals $\mathcal{P}_{\otimes_{i=1}^N \Theta}$. The joint distribution captures dependencies, while the product of marginals does not. Therefore, $H_\alpha(\mathcal{P}_{\Theta^N}\|\mathcal{P}_{\otimes_{i=1}^N \Theta})$ measures the degree of dependency among the parameters from $N$ classifiers. As a result, ***increasing the diversity of parameters in surrogate models*** and reducing their interdependence enhances adversarial transferability. This theoretical conclusion is also supported by empirical results (Li et al., 2020; Tang et al., 2024), which also advocate for generating adversarial examples from diverse models.

**The trade-off between complexity and diversity.** Reducing model complexity may conflict with increasing diversity. We discuss this issue from two angles. On one hand, when generating adversarial examples from simpler models to attack more complex ones, the overall model complex-

ity is lower, but diversity may also be limited due to the simpler structure of the ensemble attackers. On the other hand, attacking simpler models with a stronger, more diverse ensemble may increase diversity but also raise model complexity. In this scenario, reducing complexity can help prevent overfitting and lead to a tighter transferability error bound, albeit with a slight reduction in ensemble diversity. In summary, striking a balance between model complexity and diversity is crucial in practice.

**From generalization error to transferability error.** The mathematical form of Eq. (11) is in line with the generalization error bound (Bartlett & Mendelson, 2002). However, we note that a key distinction between transferability error and generalization error lies in the *independence assumption*. Conventional generalization error analysis relies on an assumption: each data point from the dataset is independently sampled (Zou & Liu, 2023; Hu et al., 2023). By contrast, the surrogate models for ensemble attack are usually trained on the datasets with similar tasks, e.g., image classification. In this case, *we cannot assume these surrogate models behave independently for a solid theoretical analysis*. To build the gap between generalization error and transferability error, our proof introduces the latest techniques in information theory (Esposito & Mondelli, 2024). And refer to Appendix E.4 for a detailed discussion about it. Thus, equipped with Theorem 1 from Esposito & Mondelli (2024), we swap the role of the model and data in learning theory literature (Geman et al., 1992; Golowich et al., 2018; Bartlett & Mendelson, 2002; Ortega et al., 2022) with analogical proof steps and prove the results.

## 4.3 The Analogy between Generalization and Adversarial Transferability

In addition to providing inspiration for model ensemble attacks, the theoretical evidence in this paper also offers new insights into another fascinating idea. Within the extensive body of research on transferable adversarial attack algorithms accumulated over the years (Gu et al., 2024), we revisit a foundational analogy that is universally applicable in the adversarial transferability literature: ***The transferability of an adversarial example is an analogue to the generalizability of the model*** (Dong et al., 2018). In other words, the ideas that enhance model generalization in deep learning may also improve adversarial transferability (Lin et al., 2019). Over the past few years, this analogy has significantly inspired the development of numerous effective algorithms, which directly reference it in their papers (Lin et al., 2019; Wang et al., 2021; Wang & He, 2021; Xiong et al., 2022; Chen et al., 2024b). And some recent papers are also inspired by it (Chen et al., 2023; Wu et al., 2024; Wang et al., 2024; Tang et al., 2024). Thus, validating this influential analogy is indispensable for defining the future landscape of adversarial transferability. Interestingly, our

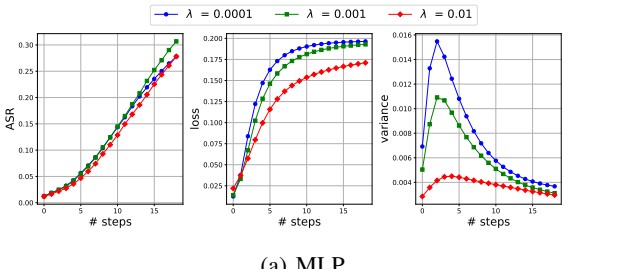 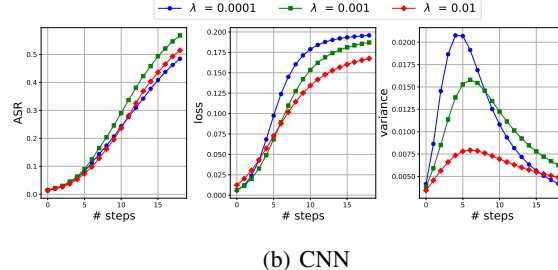

(a) MLP           (b) CNN

*Figure 2.* Evaluation of ensemble attacks with increasing the number of steps using MLPs and CNNs on the MNIST dataset.

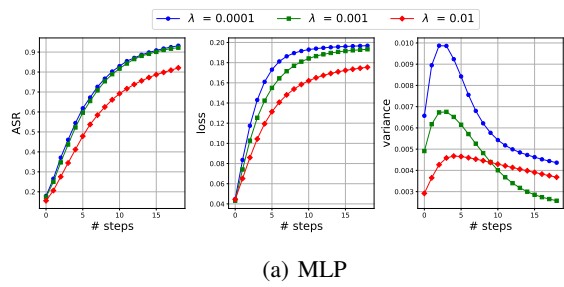 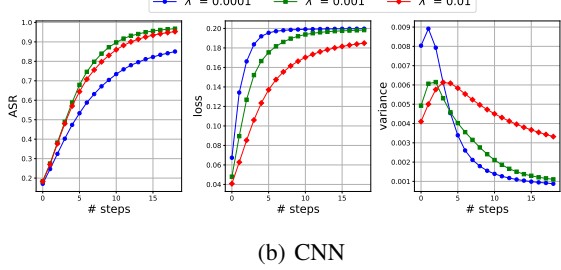

(a) MLP           (b) CNN

*Figure 3.* Evaluation of ensemble attacks with increasing the number of steps using MLPs and CNNs on the Fashion-MNIST dataset.

paper sheds light on this insight in several ways.

First, the mathematical formulations in Lemma 3.2 is similar to generalization error (Vapnik, 1998; Bousquet & Elisseeff, 2002) , which also derives an objective as a difference between the population risk and the empirical risk. Such similarity between transferability error and generalization error suggests the possible validity of the analogy. Also, Lemma 4.2 is similar to the bound of the original Rademacher complexity (Golowich et al., 2018), which also suggests that obtaining a larger training set as well as a less complex model contribute a tighter bound of Rademacher complexity. Such similarities between transferability error and generalization error suggests the possible validity of the analogy. More importantly, if the analogy is correct, then recall that in the conventional framework of learning theory: (1) increasing the size of training set typically leads to a better generalization of the model (Bousquet & Elisseeff, 2002); (2) improving the diversity among ensemble classifiers makes it more advantageous for better generalization (Ortega et al., 2022); and (3) reducing the model complexity (Cherkassky, 2002) benefits the generalization ability. It is natural to ask: in model ensemble attack, do (1) incorporating more surrogate models, (2) making them more diverse, and (3) reducing their model complexity theoretically result in better adversarial transferability?

In this section, our theoretical framework provides consistently affirmative responses to the above question as well as the analogy. Considering a higher perspective, the theory is also instructive in two ways. On the one hand, from the

perspective of a theoretical researcher, the extensive and advanced generalization theory may yield enlightening insights in the field of adversarial transferability. On the other hand, from an practitioner's point of view, ideas from deep learning algorithms can also be leveraged to develop more effective transferable attack algorithms.

## 5 Experiments

We conduct our experiments on three datasets, including the MNIST (LeCun, 1998), Fashion-MNIST (Xiao et al., 2017), and CIFAR-10 (Krizhevsky et al., 2009) datasets. We use these datasets to empirically validate our theory and build a powerful ensemble adversarial attack in practice.

We build six deep neural networks for image classification, including three MLPs with one to three hidden layers followed by a linear classification layer, and three convolutional neural networks (CNNs) with one to three convolutional layers followed by a linear classification layer. To ensure diversity among the models, we apply three different types of transformations during training. Additionally, we set the weight decay under the $L_2$ norm to $10^{-4}, 10^{-3}, 10^{-2}$, respectively. This results in a total of $6 \times 3 \times 3 = 54$ models. To establish a gold standard for adversarial transferability evaluation, we additionally train a ResNet-18 (He et al., 2016) from scratch on three datasets (MNIST, Fashion-MNIST, and CIFAR-10), respectively. We will leverage the models at hand to attack this ResNet-18 for a reliable evaluation. For models trained on MNIST, Fashion-MNIST, we set the number of epochs as 10. For

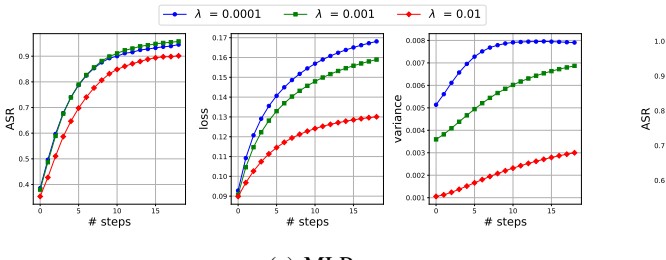
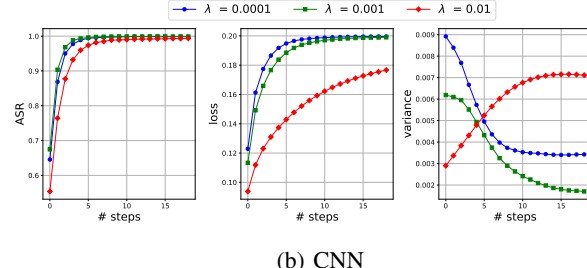

(a) MLP                    (b) CNN

*Figure 4.* Evaluation of ensemble attacks with increasing the number of steps using MLPs and CNNs on the CIFAR-10 dataset.

models trained on CIFAR-10, we set the number of epochs as 30. We use the Adam optimizer with setting the learning rate as $10^{-3}$. We set the batch size as 64.

### 5.1 Evaluation on the Attack Dynamics

For each dataset (MNIST & Fashion-MNIST & CIFAR-10), we record the attack success rate (ASR), loss value, and the variance of model predictions with increasing the number of steps for attack. We use MI-FGSM (Dong et al., 2018) to craft the adversarial example and use the cross-entropy as the loss function to optimize the adversarial perturbation. Generally, the number of steps for the transferable adversarial attack is set as 10 (Zhang et al., 2024b), but to study the attack dynamics more comprehensively, we perform 20-step attack. In our plots, we use the mean-squared-error to validate our theory, which indicates the vulnerability from the theory perspective better. The first metric exhibits an inverse relationship with transferability error. And the latter two metrics correspond to the vulnerability and diversity components in the decomposition in Section 4.1. The number of steps for attack is indicated by the $x$-axis. And we denote $\lambda$ as the weight decay. We respectively report the results on three datasets in Figure 2, Figure 3, and Figure 4.

**Vulnerability-diversity decomposition.** Across all three datasets, we observe a consistent pattern: as the number of steps increases, both ASR and loss values improve steadily, meaning that transferability error decreases while vulnerability increases. Notably, the magnitude of variance is approximately ten times smaller than that of the loss value, indicating a much smaller impact on transferability error. Thus, "vulnerability" predominantly drives the vulnerability-diversity decomposition, and the upward trend in vulnerability aligns with the reduction in transferability error.

**The trend of variance.** On the MNIST and Fashion-MNIST datasets, diversity initially increases but later declines. In contrast, on the CIFAR dataset, the variance for MLP consistently increases, whereas for CNNs, it decreases with a small regularization term but increases with a larger one. This intriguing phenomenon is tied not only to the trade-off between complexity and diversity discussed in

Section 4.2, but also to the complex behavior of variance. In the bias-variance trade-off literature (Yang et al., 2020; Lin & Dobriban, 2021; Derumigny & Schmidt-Hieber, 2023; Chen et al., 2024c), different trends in variance have been observed. For example, Yang et al. (2020) suggests that variance may follow a bell-shaped curve, rising initially and then falling as network width expands. While a full investigation of variance behavior is beyond the scope of this work, more discussion is provided in Appendix E.5.

**The potential trade-off between diversity and complexity.** Our experimental results (specifically the "variance" sub-figure), indicate the potential trade-off between diversity and complexity. Consider two distinct phases in the attack dynamics: 1) Initial phase of the attack (first few steps): During this phase, the adversarial example struggles to attack the model ensemble effectively (a low loss). Consequently, both the loss and variance increase, aligning with the vulnerability-diversity decomposition. 2) Potential "over-fitting" phase of the attack (subsequent steps): In this phase, the adversarial example can effectively attack the model ensemble, achieving a high loss. Here, the trade-off between diversity and complexity becomes evident, particularly at the final step of the attack. As the regularization term $\lambda$ increases (i.e., lower model complexity), the variance of the model ensemble may increase. For instance, in the variance sub-figure, the red curve may exceed one of the other curves, indicating this potential trade-off.

**Additional experiments.** Firstly, in Appendix D.1, we present additional experimental results on the CIFAR-100 (Krizhevsky et al., 2009) to reinforce the validity of vulnerability-diversity decomposition. Secondly, in Appendix D.2, we introduce weight norm constraints and investigate how model complexity influences ensemble complexity to support Lemma 4.2. Finally, in Appendix D.3, we use the ImageNet dataset (Russakovsky et al., 2015) to provide a straightforward demonstration of how controlling model complexity enhances adversarial transferability.

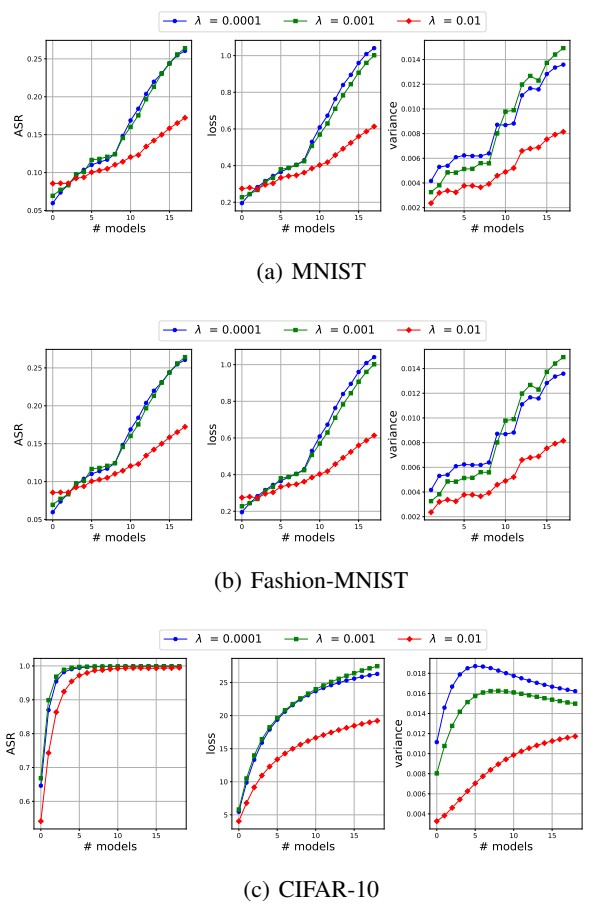

(a) MNIST

(b) Fashion-MNIST

(c) CIFAR-10

*Figure 5.* Evaluation of ensemble attacks with increasing the number of models using MLPs and CNNs on the three datasets.

### 5.2 Evaluation on the Ensemble Framework

We further validate the effectiveness of the vulnerability-diversity decomposition within the ensemble framework. Specifically, instead of focusing solely on the training dynamics, we progressively increase the number of models in the ensemble attack to evaluate the decomposition's impact. We begin by incorporating MLPs with different architectures and regularization terms, followed by CNNs. In total, up to 18 models are included in a single attack. We depicted the results in Figure 5.

We can consistently observe that increasing the number of ensemble models improves the attack success rate, i.e., reduces the transferability error. On the MNIST and Fashion-MNIST datasets, both vulnerability and diversity also increase as the number of models grows. Although the diversity sometimes shows a decreasing trend on the CIFAR-10 dataset, its magnitude is approximately 100 times smaller than vulnerability, thus having a minimal impact on ASR.

## 6 Conclusion

This paper establishes a theoretical foundation for transferable model ensemble adversarial attacks. We introduce three key concepts: transferability error, prediction variance, and empirical model ensemble Rademacher complexity. By decomposing transferability error into vulnerability and diversity, we reveal a fundamental trade-off between them. Leveraging recent mathematical tools, we derive an upper bound on transferability error, validating practical insights for enhancing adversarial transferability. Extensive experiments support our findings, advancing the understanding of transferable model ensemble adversarial attacks.

## Acknowledgement

We are deeply grateful to Bowei Zhu, Xiaolin Hu, Shaojie Li, anonymous reviewers, area chair and senior area chair for their valuable suggestions and detailed discussion. Wei Yao, Huayi Tang and Yong Liu were supported by National Natural Science Foundation of China (No.62476277), National Key Research and Development Program of China(NO. 2024YFE0203200), CCF-ALIMAMA TECH Kangaroo Fund(No.CCF-ALIMAMA OF 2024008), and Huawei-Renmin University joint program on Information Retrieval. We also acknowledge the support provided by the fund for building worldclass universities (disciplines) of Renmin University of China and by the funds from Beijing Key Laboratory of Big Data Management and Analysis Methods, Gaoling School of Artificial Intelligence, Renmin University of China, from Engineering Research Center of Next-Generation Intelligent Search and Recommendation, Ministry of Education, from Intelligent Social Governance Interdisciplinary Platform, Major Innovation & Planning Interdisciplinary Platform for the "DoubleFirst Class" Initiative, Renmin University of China, from Public Policy and Decision-making Research Lab of Renmin University of China, and from Public Computing Cloud, Renmin University of China.

## Impact Statement

We recognize the potential societal impact of our work on transferable adversarial attacks and emphasize its contribution to improving the robustness and security of machine learning models. Our study upholds high standards of scientific excellence through transparency, rigor, and reproducibility. No human subjects were involved, and no privacy or confidentiality concerns arise from the data used. We have also ensured that our work does not introduce discriminatory biases, and we are committed to the fair and inclusive participation of all individuals in the research community. While our research focuses on theoretical advancements, we are aware of the potential risks associated with adversarial

attack techniques. We encourage responsible use of these insights to build more secure AI systems and minimize any unintended harm.

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

# A    More Related Work

## A.1    Transferable Adversarial Attack

**Input transformation.**    Input transformation-based attacks have shown great effectiveness in improving transferability and can be combined with gradient-based attacks. Most input transformation techniques rely on the fundamental idea of applying data augmentation strategies to prevent overfitting to the surrogate model (Gu et al., 2024). Such methods adopt various input transformations to further improve the transferability of adversarial examples (Wang et al., 2023b;a). For instance, random resizing and padding (Xie et al., 2019), downscaling (Lin et al., 2019), mixing (Wang et al., 2021), automated data augmentation (Yan et al., 2023), block shuffle and rotation (Wang et al., 2024), and dynamical transformation (Zhu et al., 2024).

**Gradient-based optimization.**    The central concept of these methods is to develop optimization techniques in the generation of adversarial examples to achieve better transferability. Dong et al. (2018); Lin et al. (2019); Wang & He (2021) draw an analogy between generating adversarial examples and the model training process. Therefore, conventional optimization methods that improve model generalization can also benefit adversarial transferability. In gradient-based optimization methods, adversarial perturbations are directly optimized based on one or more surrogate models during inference. Some popular ideas include applying momentum (Dong et al., 2018), Nesterov accelerated gradient (Lin et al., 2019), scheduled step size (Gao et al., 2020) and gradient variance reduction (Wang & He, 2021; Xiong et al., 2022). There are also other elegantly designed techniques in recent years (Gubri et al., 2022b; Wang et al., 2022; Xiaosen et al., 2023; Li et al., 2024; Wu et al., 2024; Zhang et al., 2024b), such as collecting weights (Gubri et al., 2022b), modifying gradient calculation (Xiaosen et al., 2023) and applying integrated gradients (Ma et al., 2023).

**Model ensemble attack.**    Motivated by the use of model ensembles in machine learning, researchers have developed diverse ensemble attack strategies to obtain transferable adversarial examples (Gu et al., 2024). It is a powerful attack that employs an ensemble of models to simultaneously generate adversarial samples. It can not only integrate with advanced gradient-based optimization methods, but also harness the unique strengths of each individual model (Tang et al., 2024). Some popular ensemble paradigms include loss-based ensemble (Dong et al., 2018), prediction-based (Liu et al., 2017), logit-based ensemble (Dong et al., 2018), and longitudinal strategy (Li et al., 2020). There is also some deep analysis to compare these ensemble paradigms (Zhang et al., 2024b). Moreover, advanced ensemble algorithms have been created to ensure better adversarial transferability (Zou et al., 2020; Gubri et al., 2022a; Xiong et al., 2022; Chen et al., 2023; Li et al., 2023; Wu et al., 2024; Chen et al., 2024b).

## A.2    Statistical Learning Theory

Statistical learning theory forms the theoretical backbone of modern machine learning by providing rigorous frameworks for understanding model generalization (Vapnik, 1999). It introduces foundational concepts such as Rademacher complexity (Bartlett & Mendelson, 2002), VC dimension (Vapnik & Chervonenkis, 1971), structural risk minimization (Vapnik, 1998) . It has also been instrumental in the development of Support Vector Machines (Cortes & Vapnik, 1995) and kernel methods (Shawe-Taylor & Cristianini, 2004), which remain pivotal in supervised learning tasks. Recent advances extend statistical learning theory to deep learning, addressing challenges of high-dimensional data and model complexity (Bartlett et al., 2021). These contributions have significantly enhanced the capability to design robust learning algorithms that generalize well across diverse applications (Du & Swamy, 2013). In addition, there are also some other novel theoretical frameworks, such as information-theoretic analysis (Xu & Raginsky, 2017), PAC-Bayes bounds (Parrado-Hernández et al., 2012), transductive learning (Vapnik, 2006), and stability analysis (Bousquet & Elisseeff, 2002; Shalev-Shwartz et al., 2010). Most of them derive a bound of the order $\mathcal{O}(\frac{1}{\sqrt{M}})$, while some others derive sharper bound of generalization (Li & Liu, 2021) of the order $\mathcal{O}(\frac{1}{M})$. Such theoretical analysis suggests that with the increase of the dataset volume, the model generalization will become better.

# B Proof of Generalized Rademacher Complexity

## B.1 Preliminary

For simplicity, denote $f(\theta_i; x)$ as $f_i(x)$. For 1-Lipschitz loss function $\ell(yf(x))$ (for example, hinge loss $\ell(f(x), y) = \max(0, 1 - yf(x))$), there holds:

$$
\begin{aligned}
\mathcal{R}_N(\mathcal{Z}) &= \mathbb{E}_{\boldsymbol{\sigma}} \left[ \sup_{z \in \mathcal{Z}} \frac{1}{N} \sum_{i=1}^{N} \sigma_i \ell(f_i(x), y) \right] \\
&\leq \mathbb{E}_{\boldsymbol{\sigma}} \left[ \sup_{z \in \mathcal{Z}} \frac{1}{N} \sum_{i=1}^{N} \sigma_i y f_i(x) \right] \\
&= \mathbb{E}_{\boldsymbol{\sigma}} \left[ \sup_{z \in \mathcal{Z}} \frac{1}{N} \sum_{i=1}^{N} \sigma_i f_i(x) \right] := \Re_N(\mathcal{Z}).
\end{aligned}
$$

So we can bound $\Re_N(\mathcal{Z})$ instead of $\mathcal{R}_N(\mathcal{Z})$.

## B.2 Linear Model

Given Section B.1, we provide the bound below.

**Lemma B.1** (Linear Model). *Let $\mathcal{H} = \{x \mapsto w^T x\}$, where $x, w \in \mathbb{R}^d$. Given $N$ classifiers from $\mathcal{H}$, assume that $\|x\|_2 \leq B$ and $\|w\|_2 \leq C$. Then $\Re_N(\mathcal{Z}) \leq \frac{BC}{\sqrt{N}}$.*

*Proof.*

$$
\begin{aligned}
\Re_N(\mathcal{Z}) &= \mathbb{E}_{\boldsymbol{\sigma}} \left[ \sup_{\|x\|_2 \leq B} \frac{1}{N} \sum_{i=1}^{N} \sigma_i f_i(x) \right] \\
&= \mathbb{E}_{\boldsymbol{\sigma}} \left[ \sup_{\|x\|_2 \leq B} \frac{1}{N} \sum_{i=1}^{N} \sigma_i w_i^T x \right] && (f_i(x) = w_i^T x) \\
&= \mathbb{E}_{\boldsymbol{\sigma}} \left[ \sup_{\|x\|_2 \leq B} x^T \left( \frac{1}{N} \sum_{i=1}^{N} \sigma_i w_i \right) \right] && (a^T b = b^T a) \\
&= \frac{B}{N} \mathbb{E}_{\boldsymbol{\sigma}} \left\| \sum_{i=1}^{N} \sigma_i w_i \right\|_2 && (a^T b \leq \|a\|_2 \|b\|_2) \\
&\leq \frac{B}{N} \left( \mathbb{E}_{\boldsymbol{\sigma}} \left\| \sum_{i=1}^{N} \sigma_i w_i \right\|_2^2 \right)^{\frac{1}{2}} && \text{(Jensen inequality: } \mathbb{E}x \leq \sqrt{\mathbb{E}x^2}) \\
&= \frac{B}{N} \left\{ \mathbb{E}_{\boldsymbol{\sigma}} \left[ \left( \sum_{i=1}^{N} \sigma_i w_i^T \right) \left( \sum_{i=1}^{N} \sigma_i w_i \right) \right] \right\}^{\frac{1}{2}} \\
&= \frac{B}{N} \left[ \mathbb{E}_{\boldsymbol{\sigma}} \left( \sum_{i=1}^{N} \underbrace{\sigma_i^2}_{1} w_i^T w_i + \underbrace{\sum_{i=1}^{N} \sum_{j=1, j \neq i}^{N} \sigma_i \sigma_j w_i^T w_j}_{0} \right) \right]^{\frac{1}{2}} \\
&= \frac{B}{N} \left( \sum_{i=1}^{N} w_i^T w_i \right)^{\frac{1}{2}} \\
&\leq \frac{B}{N} \left( N \max \|w\|_2^2 \right)^{\frac{1}{2}}
\end{aligned}
$$

$$\leq \frac{BC}{\sqrt{N}}. \qquad\qquad (\|w\|_2 \leq C)$$

$\square$

## B.3 Two-layer Neural Network

Given Section B.1, we provide the bound below.

**Lemma B.2** (Two-layer Neural Network). *Let* $\mathcal{H} = \{x \mapsto w^T \phi(Ux)\}$, *where* $x \in \mathbb{R}^d$, $U \in \mathbb{R}^{m \times d}$, $w \in \mathbb{R}^m$, $m$ *is the number of the hidden layer, and* $\phi(x) = \max(0, x)$ *is the element-wise ReLU function. Given* $N$ *classifiers from* $\mathcal{H}$, *assume that* $\|x\|_2 \leq B$, $\|w\|_2 \leq B'$, *and* $\|U_i\|_2 \leq C$, *where* $U_j$ *is the j-th row of* $U$. *Then* $\Re_N(\mathcal{Z}) \leq \frac{\sqrt{m}BB'C}{\sqrt{N}}$.

*Proof.*

$$\Re_N(\mathcal{Z}) = \mathbb{E}_{\boldsymbol{\sigma}}\left[\sup_{\|x\|_2 \leq B} \frac{1}{N} \sum_{i=1}^{N} \sigma_i f_i(x)\right]$$

$$= \mathbb{E}_{\boldsymbol{\sigma}}\left[\sup_{\|x\|_2 \leq B} \frac{1}{N} \sum_{i=1}^{N} \sigma_i w_i^T \phi(U_i x)\right] \qquad (f_i(x) = w_i^T \phi(U_i x))$$

$$= \frac{B'}{N} \mathbb{E}_{\boldsymbol{\sigma}}\left[\sup_{\|x\|_2 \leq B} \left\|\sum_{i=1}^{N} \sigma_i \phi(U_i x)\right\|_2\right] \qquad (\|w\|_2 \leq B')$$

$$= \frac{B'}{N} \mathbb{E}_{\boldsymbol{\sigma}}\left[\sup_{\|x\|_2 \leq B} \left\|\sum_{i=1}^{N} \sigma_i V_i\right\|_2\right] \qquad \left(\text{Denote } V_i = \begin{bmatrix} \phi(U_{1i}x) \\ \vdots \\ \phi(U_{mi}x) \end{bmatrix} \in \mathbb{R}^m\right)$$

$$= \frac{B'}{N} \mathbb{E}_{\boldsymbol{\sigma}}\left[\sup_{\|x\|_2 \leq B} \sqrt{\left(\sum_{i=1}^{N} \sigma_i V_i^T\right)\left(\sum_{i=1}^{N} \sigma_i V_i\right)}\right]$$

$$= \frac{B'}{N} \mathbb{E}_{\boldsymbol{\sigma}}\left[\sup_{\|x\|_2 \leq B} \left(\sum_{i=1}^{N} \underbrace{\sigma_i^2}_{1} V_i^T V_i + \underbrace{\sum_{i=1}^{N} \sum_{j=1, j\neq i}^{N} \sigma_i \sigma_j V_i^T V_j}_{0}\right)^{\frac{1}{2}}\right]$$

$$= \frac{B'}{N} \sup_{\|x\|_2 \leq B} \left(\sum_{i=1}^{N} V_i^T V_i\right)^{\frac{1}{2}}$$

$$\leq \frac{B'}{N} \sup_{\|x\|_2 \leq B} \left(N \max_i \|V_i\|_2^2\right)^{\frac{1}{2}}$$

$$\leq \frac{B'}{\sqrt{N}} \sup_{\|x\|_2 \leq B} \left(\max_i \|V_i\|_2\right)$$

For $V_i = \begin{bmatrix} \phi(U_{1i}x) \\ \vdots \\ \phi(U_{mi}x) \end{bmatrix} \in \mathbb{R}^m$, we have

$$\sup_{\|x\|_2 \leq B} \left(\max_i \|V_i\|_2\right) = \sup_{\|x\|_2 \leq B} \left(\max_i \left\|\begin{bmatrix} \phi(U_{1i}x) \\ \vdots \\ \phi(U_{mi}x) \end{bmatrix}\right\|_2\right)$$

$$\leq \sup_{\|x\|_2 \leq B} \left( \max_i \left\| \begin{bmatrix} U_{1i}x \\ \vdots \\ U_{mi}x \end{bmatrix} \right\|_2 \right) \qquad (|\phi(x)| \leq |x|)$$

$$\leq \sqrt{m} \sup_{\|x\|_2 \leq B} \left( \max_i \max_j \|U_{ji}x\|_2 \right)$$

$$\leq \sqrt{m} \sup_{\|x\|_2 \leq B} \left( \max_i \max_j \|U_{ji}\|_2 \|x\|_2 \right)$$

$$= \sqrt{m}BC \qquad (\|x\|_2 \leq B \text{ and } \|U_{ji}\|_2 \leq C)$$

Finally,

$$\Re_N(\mathcal{Z}) \leq \frac{B'}{\sqrt{N}} \sup_{\|x\|_2 \leq B} \left( \max_i \|V_i\|_2 \right) \leq \frac{\sqrt{m}BB'C}{\sqrt{N}}$$

The proof is complete.

$\square$

## B.4 Proof of Lemma 4.2

For simplicity, denote $f(\theta_i; x)$ as $f_i(x)$ and $i \in \{1, \cdots, N\}$ as $i \in [N]$.

First, we begin with a lemma, which is a similar version of Lemma 1 from (Golowich et al., 2018).

**Lemma B.3.** *Let $\phi$ be a 1-Lipschitz, positive-homogeneous activation function which is applied element-wise (such as the ReLU). Then for any class of vector-valued functions $\mathcal{F}$, any convex and monotonically increasing function $g : \mathbb{R} \to [0, \infty)$ and $R \in \mathbb{R}_+$, there holds:*

$$\mathbb{E}_{\boldsymbol{\sigma}} \sup_{f \in \mathcal{F}, W: \|W\|_F \leq R} g\left( \left\| \sum_{i=1}^N \sigma_i \phi\left(W f_i(x)\right) \right\| \right) \leq 2 \cdot \mathbb{E}_{\boldsymbol{\sigma}} \sup_{f \in \mathcal{F}} g\left( R \cdot \left\| \sum_{i=1}^N \sigma_i f_i(x) \right\| \right) \qquad (12)$$

*Proof.* Let $w_1, \cdots, w_h$ be the rows of $W$, we have

$$\left\| \sum_{i=1}^N \sigma_i \phi\left(W f_i(x)\right) \right\|^2 = \sum_{j=1}^h \left[ \sum_{i=1}^N \sigma_i \phi(w_j f_i(x)) \right]^2$$

$$= \sum_{j=1}^h \|w_j\|^2 \left[ \sum_{i=1}^N \sigma_i \phi\left( \frac{w_j^\top}{\|w_j\|} f_i(x) \right) \right]^2 \qquad (\phi(ax) = a\phi(x))$$

Therefore, the supremum of this over all $w_1, \cdots, w_h$ such that $\|W\|_F^2 = \sum_{j=1}^h \|w_j\|^2 \leq R^2$ must be attained when $\|w_j\| = R$ for some $j$ and $\|w_i\| = 0$ for all $i \neq j$. So we have

$$\mathbb{E}_{\boldsymbol{\sigma}} \sup_{f \in \mathcal{F}, W: \|W\|_F \leq R} g\left( \left\| \sum_{i=1}^N \sigma_i \phi\left(W f_i(x)\right) \right\| \right) = \mathbb{E}_{\boldsymbol{\sigma}} \sup_{f \in \mathcal{F}, w: \|w\| = R} g\left( \left\| \sum_{i=1}^N \sigma_i \phi\left(w^\top f_i(x)\right) \right\| \right).$$

Since $g(|z|) \leq g(z) + g(-z)$, this can be upper bounded by

$$\mathbb{E}_{\boldsymbol{\sigma}} \sup g\left( \sum_{i=1}^N \sigma_i \phi\left(w^\top f_i(x)\right) \right) + \mathbb{E}_{\boldsymbol{\sigma}} \sup g\left( -\sum_{i=1}^N \sigma_i \phi\left(w^\top f_i(x)\right) \right) = 2 \cdot \mathbb{E}_{\boldsymbol{\sigma}} \sup g\left( \sum_{i=1}^N \sigma_i \phi\left(w^\top f_i(x)\right) \right),$$

where the equality follows from the symmetry in the distribution of the $\sigma_i$ random variables. The right hand side in turn can be upper bounded by

$$2 \cdot \mathbb{E}_{\boldsymbol{\sigma}} \sup_{f \in \mathcal{F}, w: \|w\| = R} g\left( \sum_{i=1}^N \sigma_i w^\top f_i(x) \right) \leq 2 \cdot \mathbb{E}_{\boldsymbol{\sigma}} \sup_{f \in \mathcal{F}, w: \|w\| = R} g\left( \|w\| \left\| \sum_{i=1}^N \sigma_i f_i(x) \right\| \right)$$

$$= 2 \cdot \mathbb{E}_{\boldsymbol{\sigma}} \sup_{f \in \mathcal{F}} g \left( R \cdot \left\| \sum_{i=1}^{N} \sigma_i f_i(x) \right\| \right).$$

$\square$

With this lemma in hand, we can prove Lemma 4.2:

*Proof.* For $\lambda > 0$, the rademacher complexity can be upper bounded as

$$N \Re_N (\mathcal{Z}) = \mathbb{E}_{\boldsymbol{\sigma}} \sup_{f_1, \cdots, f_n} \sum_{i=1}^{N} \sigma_i f_i(x)$$

$$\leq \frac{1}{\lambda} \log \mathbb{E}_{\boldsymbol{\sigma}} \sup \exp \left( \lambda \sum_{i=1}^{N} \sigma_i f_i(x) \right) \qquad \text{(Jensen's inequality)}$$

$$\leq \frac{1}{\lambda} \log \mathbb{E}_{\boldsymbol{\sigma}} \sup \exp \left( \underbrace{\sup_{i \in [n]} \|W_{i,l}\|_F}_{T_l} \left\| \lambda \sum_{i=1}^{N} \sigma_i \phi_{l-1} \underbrace{(W_{i,l-1} \phi_{l-2} (\ldots \phi_1 (W_{i,1} x)))}_{f_{i,l-1}(x)} \right\| \right)$$

We write this last expression as

$$\frac{1}{\lambda} \log \mathbb{E}_{\boldsymbol{\sigma}} \sup \exp \left( T_l \cdot \lambda \left\| \sum_{i=1}^{N} \sigma_i \phi_{l-1} (f_{i,l-1}(x)) \right\| \right)$$

$$\leq \frac{1}{\lambda} \log \left( 2 \cdot \mathbb{E}_{\boldsymbol{\sigma}} \sup \exp \left( T_l \cdot T_{l-1} \cdot \lambda \left\| \sum_{i=1}^{N} \sigma_i f_{i,l-2}(x) \right\| \right) \right) \qquad \text{(Lemma B.3)}$$

$$\leq \cdots \qquad \text{(Repeatedly apply Lemma B.3)}$$

$$\leq \frac{1}{\lambda} \log \left( 2^{l-2} \cdot \mathbb{E}_{\boldsymbol{\sigma}} \sup \exp \left( \lambda \cdot \prod_{i=1}^{l-1} T_i \cdot \left\| \sum_{i=1}^{N} \sigma_i \phi_1 (W_{i,1} x) \right\| \right) \right)$$

$$\leq \frac{1}{\lambda} \log \left( 2^{l-1} \cdot \mathbb{E}_{\boldsymbol{\sigma}} \sup \exp \left( \lambda \cdot \prod_{i=1}^{l-1} T_i \cdot \left\| \sum_{i=1}^{N} \sigma_i W_{i,1} x \right\| \right) \right)$$

Assume that $W_{i,1}^*, i \in [N]$ maximizes

$$\sup \exp \left( \lambda \cdot \prod_{i=1}^{l-1} T_i \cdot \left\| \sum_{i=1}^{N} \sigma_i W_{i,1} x \right\| \right).$$

Therefore,

$$\frac{1}{\lambda} \log \left( 2^{l-1} \cdot \mathbb{E}_{\boldsymbol{\sigma}} \sup \exp \left( \lambda \cdot \prod_{i=1}^{l-1} T_i \cdot \left\| \sum_{i=1}^{N} \sigma_i W_{i,1} x \right\| \right) \right)$$

$$= \frac{1}{\lambda} \log \left( 2^{l-1} \cdot \mathbb{E}_{\boldsymbol{\sigma}} \exp \left( \lambda \cdot \underbrace{\prod_{i=1}^{l-1} T_i \cdot \left\| \sum_{i=1}^{N} \sigma_i W_{i,1}^* x \right\|}_{Z} \right) \right)$$

$$= \frac{1}{\lambda} \log \left( 2^{l-1} \cdot \mathbb{E}_{\boldsymbol{\sigma}} \exp (\lambda Z) \right)$$

$$= \frac{(l-1)\log(2)}{\lambda} + \frac{1}{\lambda}\log\left\{\mathbb{E}_{\boldsymbol{\sigma}}\exp\left(\lambda Z\right)\right\}$$

$$= \frac{(l-1)\log(2)}{\lambda} + \frac{1}{\lambda}\log\{\mathbb{E}\exp\lambda(Z - \mathbb{E}Z)\} + \mathbb{E}Z$$

For $\mathbb{E}Z$, we have

$$\mathbb{E}Z = \prod_{i=1}^{l-1} T_i \sqrt{\mathbb{E}_{\boldsymbol{\sigma}}\left[\left\|\sum_{i=1}^{N}\sigma_i W_{i,1}^* x\right\|^2\right]}$$

$$= \prod_{i=1}^{l-1} T_i \sqrt{\mathbb{E}_{\boldsymbol{\sigma}}\left[\sum_{i=j}^{N}\sigma_i\sigma_j \left(W_{i,1}^* x\right)^T \left(W_{j,1}^* x\right)\right]}$$

$$\leq \prod_{i=1}^{l-1} T_i \left(T_1 B\sqrt{N}\right)$$

$$= B\sqrt{N}\prod_{i=1}^{l} T_i$$

Note that $Z$ is a deterministic function of the *i.i.d.* random variables $\sigma_1, \cdots, \sigma_N$, and satisfies

$$Z(\sigma_1, \cdots, \sigma_i, \cdots, \sigma_N) - Z(\sigma_1, \cdots, -\sigma_i, \cdots, \sigma_N) \leq 2B\underbrace{\prod_{i=1}^{l} T_i}_{T}.$$

This means that $Z$ satisfies a bounded-difference condition. According to Theorem 6.2 in Boucheron et al. (2013), $Z$ is sub-Gaussian with variance factor

$$\frac{1}{4}\sum_{i=1}^{N}(2BT)^2 = NB^2T^2,$$

and satisfies

$$\frac{1}{\lambda}\log\{\mathbb{E}\exp\lambda(Z - \mathbb{E}Z)\} \leq \frac{1}{\lambda}\cdot\frac{\lambda^2}{2}NB^2T^2 = \frac{\lambda}{2}NB^2T^2.$$

Choosing $\lambda = \frac{\sqrt{2\log(2)l}}{BT\sqrt{N}}$ and using the above, we get that

$$\frac{(l-1)\log(2)}{\lambda} + \frac{1}{\lambda}\log\{\mathbb{E}\exp\lambda(Z - \mathbb{E}Z)\} + \mathbb{E}Z \leq \left(\sqrt{(2\log 2)l} + 1\right)BT\sqrt{N}$$

Finally, we get

$$\Re_N\left(\mathcal{Z}\right) \leq \frac{\left(\sqrt{(2\log 2)l} + 1\right)BT}{\sqrt{N}}$$

$\square$

## C   Main Proof

### C.1   Transferability Error and Generalization Error

For $z = (x, y)$, there holds

$$TE(z) = L_P(z^*) - L_P(z) \leq L_P(z^*) - L_P(z) + (L_E(z) - L_E(z^*))$$

$$= (L_P(z^*) - L_E(z^*)) + (L_E(z) - L_P(z))$$
$$\leq \sup_{x \in \mathcal{B}_\epsilon(x)} (L_P(z) - L_E(z)) + \sup_{x \in \mathcal{B}_\epsilon(x)} (L_E(z) - L_P(z))$$
$$\leq \sup_{z \in \mathcal{Z}} (L_P(z) - L_E(z)) + \sup_{z \in \mathcal{Z}} (L_E(z) - L_P(z)).$$
$$\leq 2 \sup_{z \in \mathcal{Z}} |L_P(z) - L_E(z)|.$$

## C.2 Proof of Theorem 4.1

We prove a general version of the theorem as follows:

**Theorem C.1.** *Consider the squared error loss* $l(\theta, x, y) = [f(\theta; x) - y]^2$ *for a data point* $z = (x, y)$. *Assume that the data is generated by a function* $g(x)$ *such that* $y = g(x) + \rho$, *where the zero-mean noise* $\rho$ *has a variance of* $\eta^2$ *and is independent of* $x$. *Then there holds*

$$TE(z, \epsilon) = L_P(z^*) - \eta^2 - \underbrace{Var_{\theta \sim \mathcal{P}_\Theta} f(\theta; x)}_{Diversity} - \underbrace{[g(x) - \mathbb{E}_{\theta \sim \mathcal{P}_\Theta} f(\theta; x)]^2}_{Attack}. \tag{13}$$

*Remark.* The irreducible error $\eta^2$ is constant because it arises from inherent noise and randomness in the data (Geman et al., 1992).

*Proof.* Given Eq. (5), it is equivalent to prove

$$L_P(z) = Var_\theta f(\theta; x) + [g(x) - \mathbb{E}_{\theta \sim \mathcal{P}_\Theta} f(\theta; x)]^2 + \eta^2. \tag{14}$$

Note that

$$L_P(z) = \mathbb{E}_{\theta \sim \mathcal{P}_\Theta} [f(\theta; x) - y]^2$$
$$= \mathbb{E}_{\theta \sim \mathcal{P}_\Theta} [f(\theta; x) - g(x) + g(x) - y]^2$$
$$= \mathbb{E}_{\theta \sim \mathcal{P}_\Theta} \left[ (f(\theta; x) - g(x))^2 + (g(x) - y)^2 + 2(g(x) - y)(f(\theta; x) - g(x)) \right].$$

Recall that $y = g(x) + \rho$ with $\mathbb{E}(\rho) = 0$ and $Var(\rho) = \eta^2$, we have

$$\mathbb{E}_{\theta \sim \mathcal{P}_\Theta} (g(x) - y)^2 = \eta^2,$$

and

$$\mathbb{E}_{\theta \sim \mathcal{P}_\Theta} [2(g(x) - y)(f(\theta; x) - g(x))] = -2\mathbb{E}(\rho)\mathbb{E}_{\theta \sim \mathcal{P}_\Theta} [f(\theta; x) - g(x)] = 0.$$

Therefore,

$$L_P(z) = \mathbb{E}_{\theta \sim \mathcal{P}_\Theta} [f(\theta; x) - g(x)]^2 + \eta^2. \tag{15}$$

Likewise, we decompose the first term as

$$\mathbb{E}_\theta [f(\theta; x) - g(x)]^2$$
$$= \mathbb{E}_\theta [f(\theta; x) - \mathbb{E}_\theta f(\theta; x) + \mathbb{E}_\theta f(\theta; x) - g(x)]^2$$
$$= \mathbb{E}_\theta \left[ (f(\theta; x) - \mathbb{E}_\theta f(\theta; x))^2 + (\mathbb{E}_\theta f(\theta; x) - g(x))^2 \right.$$
$$\left. - 2(f(\theta; x) - \mathbb{E}_\theta f(\theta; x))(\mathbb{E}_\theta f(\theta; x) - g(x)) \right]$$
$$= \underbrace{\mathbb{E}_\theta (f(\theta; x) - \mathbb{E}_\theta f(\theta; x))^2}_{Var_\theta f(\theta; x)} + \underbrace{\mathbb{E}_\theta (\mathbb{E}_\theta f(\theta; x) - g(x))^2}_{(g(x) - \mathbb{E}_\theta (f(\theta; x))^2}$$
$$- 2 \underbrace{\mathbb{E}_\theta [f(\theta; x) - \mathbb{E}_\theta f(\theta; x)](\mathbb{E}_\theta f(\theta; x) - g(x))}_{0},$$

with the derivations for the second and third term:

$$\mathbb{E}_\theta (f(\theta; x) - \mathbb{E}_\theta f(\theta; x))^2 = (\mathbb{E}_\theta f(\theta; x))^2 - 2g(x)\mathbb{E}_\theta f(\theta; x) + g^2(x)$$

$$= (g(x) - \mathbb{E}_\theta(f(\theta; x))^2,$$

and

$$
\begin{aligned}
&\mathbb{E}_\theta \left[ f(\theta; x) - \mathbb{E}_\theta f(\theta; x))(\mathbb{E}_\theta f(\theta; x) - g(x) \right] \\
=&(\mathbb{E}_\theta f(\theta; x))^2 - g(x)\mathbb{E}_\theta f(\theta; x) - (\mathbb{E}_\theta f(\theta; x))^2 + g(x)\mathbb{E}_\theta f(\theta; x) \\
=&0.
\end{aligned}
$$

As a result,

$$\mathbb{E}_\theta \left[ f(\theta; x) - g(x) \right]^2 = Var_\theta f(\theta; x) + [g(x) - \mathbb{E}_{\theta \sim \mathcal{P}_\Theta} f(\theta; x)]^2. \tag{16}$$

Combining the above results and we complete the proof.

$\square$

To prove Theorem 4.1, we just set $\rho = 0$ in the above general version of theorem.

Similarly, consider the empirical version of Theorem 4.1, we decompose $L_E(z)$ as follows:

**Theorem C.2** (Vulnerability-diversity Decomposition (empirical version)). *Consider the squared error loss* $l(f(\theta; x), y) = [f(\theta; x) - y]^2$ *for a data point* $z = (x, y)$. *Let* $\hat{f}(\theta; x) = \frac{1}{N} \sum_{i=1}^N f(\theta_i; x)$ *be the expectation of prediction over the distribution on the parameter space. Then there holds*

$$
\begin{aligned}
L_E(z) &= \frac{1}{N} \sum_{i=1}^N \ell(f(\theta_i; x), y) \\
&= \underbrace{l(\hat{f}(\theta; x), y)}_{Vulnerability} + \underbrace{\frac{1}{N} \sum_{j=1}^N \left( f(\theta_i; x) - \frac{1}{N} \sum_{j=1}^N f(\theta_i; x) \right)^2}_{Diversity}.
\end{aligned}
$$

The proof is similar to the above:

$$
\begin{aligned}
L_E(z) &= \frac{1}{N} \sum_{i=1}^N (f(\theta_i; x) - y)^2 \\
&= \frac{1}{N} \frac{1}{N} \sum_{i=1}^N \left( f(\theta_i; x) - \frac{1}{N} \sum_{i=1}^N f(\theta_i; x) + \frac{1}{N} \sum_{i=1}^N f(\theta_i; x) - y \right)^2 \\
&= \frac{1}{N} \sum_{i=1}^N \left[ \left( f(\theta_i; x) - \frac{1}{N} \sum_{i=1}^N f(\theta_i; x) \right)^2 + \left( \frac{1}{N} \sum_{i=1}^N f(\theta_i; x) - y \right)^2 + \right. \\
&\quad \left. 2 \left( f(\theta_i; x) - \frac{1}{N} \sum_{i=1}^N f(\theta_i; x) \right) \left( \frac{1}{N} \sum_{i=1}^N f(\theta_i; x) - y \right) \right] \\
&= \underbrace{l(\hat{f}(\theta; x), y)}_{Vulnerability} + \underbrace{\frac{1}{N} \sum_{j=1}^N \left( f(\theta_i; x) - \frac{1}{N} \sum_{j=1}^N f(\theta_i; x) \right)^2}_{Diversity} + \\
&\quad \frac{2}{N} \sum_{i=1}^N \left( f(\theta_i; x) - \frac{1}{N} \sum_{i=1}^N f(\theta_i; x) \right) \left( \frac{1}{N} \sum_{i=1}^N f(\theta_i; x) - y \right).
\end{aligned}
$$

The last terms equals to 0 because

$$\sum_{i=1}^N \left( f(\theta_i; x) - \frac{1}{N} \sum_{i=1}^N f(\theta_i; x) \right) \left( \frac{1}{N} \sum_{i=1}^N f(\theta_i; x) - y \right)$$

$$= \frac{1}{N} \left( \sum_{i=1}^{N} f(\theta_i; x) \right)^2 - y \sum_{i=1}^{N} f(\theta_i; x) - \frac{1}{N} \left( \sum_{i=1}^{N} f(\theta_i; x) \right)^2 + y \sum_{i=1}^{N} f(\theta_i; x)$$

$$= 0.$$

The proof is complete.

### C.3  Proof of Theorem 4.1 (KL Divergence Loss)

In this section, we consider a different problem setting and show how to extend Theorem 4.1 to KL divergence loss. We first define multi-class classification in the context of transferable model ensemble adversarial attack.

**Multi-class classification.**  Consider a $k$-classification problem. Given the input space $\mathcal{X} \subset \mathbb{R}^d$ and the output space $\mathcal{Y} \subset \mathbb{R}^k$, we have a joint distribution $\mathcal{P}_{\mathcal{Z}}$ over the input space $\mathcal{Z} = \mathcal{X} \times \mathcal{Y}$. The training set $Z_{\text{train}} = \{\mathbf{z}_i | \mathbf{z}_i = (\mathbf{x}_i, \mathbf{y}_i) \in \mathcal{Z}, \mathbf{y}_i \in \{0,1\}^k, \|\mathbf{y}_i\|_1 = 1, i = 1, \cdots, M\}$, which consists of $M$ examples drawn independently from $\mathcal{P}_{\mathcal{Z}}$. We denote the hypothesis space by $\mathcal{H} : \mathcal{X} \mapsto \mathcal{Y}$ and the parameter space by $\Theta$. Let $f(\boldsymbol{\theta}; \cdot) \in \mathcal{H}$ be a classifier parameterized by $\boldsymbol{\theta} \in \Theta$, trained for a classification task using a loss function $\ell : \mathcal{Y} \times \mathcal{Y} \mapsto \mathbb{R}_0^+$. Let $\mathcal{P}_{\Theta}$ represent the distribution over the parameter space $\Theta$. Define $\mathcal{P}_{\Theta^N}$ as the joint distribution over the product space $\Theta^N$, which denotes the space of $N$ such sets of parameters. We use $Z_{\text{train}}$ to train $N$ surrogate models $f(\boldsymbol{\theta}_1; \cdot), \cdots, f(\boldsymbol{\theta}_N; \cdot)$ for model ensemble. The training process of these $N$ classifiers can be viewed as sampling the parameter sets $\boldsymbol{\theta}^N = (\boldsymbol{\theta}_1, \ldots, \boldsymbol{\theta}_N)$ from the distribution $\mathcal{P}_{\Theta^N}$, i.e., $\boldsymbol{\theta}^N \sim \mathcal{P}_{\Theta^N}$. For a data point $\mathbf{z} = (\mathbf{x}, \mathbf{y}) \in \mathcal{Z}$ and $N$ classifiers for model ensemble attack, let the model output be normalized (i.e., $\|f(\boldsymbol{\theta}; \mathbf{x})\|_1 = 1$). Define the empirical risk $L_E(\mathbf{z})$ and the population risk $L_P(\mathbf{z})$ of the adversarial example $\mathbf{z}$ as

$$L_E(\mathbf{z}) = \frac{1}{N} \sum_{i=1}^{N} \ell(f(\boldsymbol{\theta}_i; \mathbf{x}), \mathbf{y}), \tag{17}$$

$$\text{and} \quad L_P(\mathbf{z}) = \mathbb{E}_{\boldsymbol{\theta}^N \sim \mathcal{P}_{\Theta^N}} L_E(\mathbf{z}). \tag{18}$$

Intuitively, a transferable adversarial example leads to a large $L_P(\mathbf{z})$ because it can attack many classifiers with parameter $\boldsymbol{\theta} \in \Theta$. Therefore, the most transferable adversarial example $\mathbf{z}^* = (\mathbf{x}^*, \mathbf{y})$ around $\mathbf{z}$ is defined as

$$\mathbf{x}^* = \arg \max_{\mathbf{x} \in \mathcal{B}_\epsilon(\mathbf{x})} L_P(\mathbf{z}), \tag{19}$$

where $\mathcal{B}_\epsilon(\mathbf{x}) = \{\hat{\mathbf{x}} : \|\hat{\mathbf{x}} - \mathbf{x}\|_2 \le \epsilon\}$ is an adversarial region centered at $\hat{\mathbf{x}}$ with radius $\epsilon > 0$. However, the expectation in $L_P(\mathbf{z})$ cannot be computed directly. Thus, when generating adversarial examples, the empirical version Eq. (2) is used in practice, such as loss-based ensemble attack (Dong et al., 2018). So the adversarial example $\mathbf{z} = (\mathbf{x}, \mathbf{y})$ is obtained from

$$\mathbf{x} = \arg \max_{\mathbf{x} \in \mathcal{B}_\epsilon(\mathbf{x})} L_E(\mathbf{z}). \tag{20}$$

There is a gap between the adversarial example $\mathbf{z}$ we find and the most transferable one $\mathbf{z}^*$. It is due to the fact that the ensemble classifiers cannot cover the whole parameter space of the classifier, i.e., there is a difference between $L_P(\mathbf{z})$ and $L_E(\mathbf{z})$. Accordingly, the core objective of transferable model ensemble attack is to design approaches that approximate $L_E(\mathbf{z})$ to $L_P(\mathbf{z})$, thereby increasing the transferability of adversarial examples.

Note that the training process of $N$ classifiers can be viewed as sampling the parameter sets $\overline{\theta}^N = (\overline{\theta}_1, \ldots, \overline{\theta}_N)$ from the distribution $\mathcal{P}_{\Theta^N}$, i.e., $\overline{\theta}^N \sim \mathcal{P}_{\Theta^N}$. We generate a transferable adversarial example using these $N$ models and evaluate its performance on another $N$ models $\theta^N = (\theta_1, \ldots, \theta_N)$, which is an independent copy of $\overline{\theta}^N$. For a data $z = (x, y) \in \mathcal{Z}$ and the parameter set $\theta^N$, our aim is to bound **the difference of attack performance between the given $N$ models $\overline{\theta}^N$ and $N$ unknown models $\theta^N$**. In other words, if

- An adversarial example $z$ can effectively attack the given model ensemble, i.e., a large $L_E(\mathbf{z})$.

- There is guarantee for the difference of attack performance between known and unknown models, i.e., a small $\left| \mathbb{E}_{\mathbf{z}, \boldsymbol{\theta}^N \sim \mathcal{P}_{\mathcal{Z}, \Theta^N}} [L_P(\mathbf{z}) - L_E(\mathbf{z})] \right|$.

Then there is adversarial transferability guarantee for $z$. We perform the decomposition to analyze $L_E(\mathbf{z})$ in this section. While we provide an information-theoretic analysis to deal with $\left| \mathbb{E}_{\mathbf{z}, \boldsymbol{\theta}^N \sim \mathcal{P}_{\mathcal{Z}, \Theta^N}} \left[ L_P(\mathbf{z}) - L_E(\mathbf{z}) \right] \right|$ in Appendix C.7.

Now we decompose $L_E(\mathbf{z})$ into vulnerability, diversity and constants. It is a similar version of Theorem 4.1 using KL divergence loss.

**Proposition C.3** (Vulnerability-diversity Decomposition). *Consider KL divergence as the loss function, i.e., $\ell(f(\boldsymbol{\theta}_i; \mathbf{x}), \mathbf{y}) = \sum_{j=1}^{k} f(\boldsymbol{\theta}_i; \mathbf{x}) \log \frac{f(\boldsymbol{\theta}_i; \mathbf{x})}{\mathbf{y}}$. Let $\overline{f}(\boldsymbol{\theta}; \mathbf{x})$ be the normalized geometric mean of ensembles $\{\mathbf{f}_i\}_{i=1}^{N}$. Then there holds*

$$L_E(\mathbf{z}) = \underbrace{\ell(\mathbf{y}, \overline{f}(\boldsymbol{\theta}; \mathbf{x}))}_{\text{Vulnerability}} + \underbrace{\frac{1}{N} \sum_{i=1}^{N} \ell(\overline{f}(\boldsymbol{\theta}; \mathbf{x}), f(\boldsymbol{\theta}_i; \mathbf{x}))}_{\text{Diversity}}. \tag{21}$$

The "Vulnerability" term measures the risk of a data point $\mathbf{z}$ being compromised by the model ensemble. If the model ensemble is sufficiently strong to fit the direction opposite to the target label, the resulting high loss theoretically improves $L_E(\mathbf{z})$. This insight suggests that *selecting strong attackers* as ensemble components leads to lower $L_E(\mathbf{z})$. The "Diversity" term implies that *selecting diverse attackers* in a model ensemble attack theoretically contributing to a increase in $L_E(\mathbf{z})$. In conclusion, it provides similar guideline comparing to Theorem 4.1: we are supposed to choose ensemble components that are both strong and diverse.

*Proof.* We first introduce Bregman divergence.

**Definition C.4** (Bregman divergence). Let $\phi : \Omega \to \mathbb{R}$ be a function that is: a) strictly convex, b) continuously differentiable, $c$ ) defined on a closed convex set $\Omega$. Then the Bregman divergence is defined as

$$B_\phi(\mathbf{x}, \mathbf{y}) = \phi(\mathbf{x}) - \phi(\mathbf{y}) - \langle \nabla \phi(\mathbf{y}), \mathbf{x} - \mathbf{y} \rangle, \quad \forall \mathbf{x}, \mathbf{y} \in \Omega.$$

That is, the difference between the value of $\phi$ at $\mathbf{x}$ and the first order Taylor expansion of $\phi$ around $\mathbf{y}$ evaluated at point $\mathbf{x}$. Notice that let $\Omega = \mathcal{Y}$ and KL divergence can be a special case of Bregman divergence if $\phi(\mathbf{x}) = \sum_i (x_i \log x_i - x_i)$ or $\phi(\mathbf{x}) = \sum_i x_i \log x_i$, where $x_i$ ($i \in 1, \cdots, k$) are the components of $\mathbf{x}$.

Now we start the proof. It follows the Bregman ambiguity decomposition in Wood et al. (2024).

Denote $\mathbf{f}_i = f(\boldsymbol{\theta}_i; \mathbf{x}) \in \mathbb{R}^k$ and

$$\overline{\mathbf{f}} = [\nabla \phi]^{-1} \left( \frac{1}{N} \sum_{i=1}^{N} \nabla \phi(\mathbf{f}_i) \right), \tag{22}$$

which is the Bregman Centroid Combiner (Wood et al., 2024) of ensembles $\{\mathbf{f}_i\}_{i=1}^{N}$. Therefore, we have

$$\nabla \phi(\overline{\mathbf{f}}) = \frac{1}{N} \sum_{i=1}^{N} \nabla \phi(\mathbf{f}_i),$$

so that

$$\frac{1}{N} \sum_{i=1}^{N} \langle \overline{\mathbf{f}} - \mathbf{y}, \nabla \phi(\mathbf{f}_i) \rangle = \langle \overline{\mathbf{f}} - \mathbf{y}, \nabla \phi(\overline{\mathbf{f}}) \rangle.$$

In other words

$$B_\phi(\mathbf{y}, \overline{\mathbf{f}}) = \phi(\mathbf{y}) - \phi(\overline{\mathbf{f}}) - \langle \mathbf{y} - \overline{\mathbf{f}}, \nabla \phi(\overline{\mathbf{f}}) \rangle$$

$$= \phi(\mathbf{y}) - \phi(\overline{\mathbf{f}}) + \frac{1}{N} \sum_{i=1}^{N} \langle \overline{\mathbf{f}} - \mathbf{y}, \nabla \phi(\mathbf{f}_i) \rangle$$

$$= \left[ \phi(\mathbf{y}) - \frac{1}{N} \sum_{i=1}^{N} \phi(\mathbf{f}_i) - \frac{1}{N} \sum_{i=1}^{N} \langle \mathbf{y} - \mathbf{f}_i, \nabla \phi(\mathbf{f}_i) \rangle \right] + \left[ \frac{1}{N} \sum_{i=1}^{N} \phi(\mathbf{f}_i) - \phi(\overline{\mathbf{f}}) + \frac{1}{N} \sum_{i=1}^{N} \langle \overline{\mathbf{f}} - \mathbf{f}_i, \nabla \phi(\mathbf{f}_i) \rangle \right]$$

$$= \frac{1}{N} \sum_{i=1}^{N} B_\phi(\mathbf{y}, \mathbf{f}_i) - \frac{1}{N} \sum_{i=1}^{N} B_\phi(\bar{\mathbf{f}}, \mathbf{f}_i). \tag{23}$$

Let $\phi(\mathbf{x}) = \sum_i (x_i \log x_i - x_i)$ in Eq. (23) and we have

$$\mathrm{D}_{\mathrm{KL}}(\mathbf{y}, \bar{\mathbf{f}}) = \underbrace{\frac{1}{N} \sum_{i=1}^{N} \mathrm{D}_{\mathrm{KL}}(\mathbf{y}, \mathbf{f}_i)}_{L_E(\mathbf{z})} - \frac{1}{N} \sum_{i=1}^{N} \mathrm{D}_{\mathrm{KL}}(\bar{\mathbf{f}}, \mathbf{f}_i).$$

Replace $\mathrm{D}_{\mathrm{KL}}$ with $\ell$ and we can prove the result. $\qquad \square$

### C.4 Proof of Theorem 4.3

We first define a divergence measure taken into account. Given a measurable space and two measures $\mu, \nu$ which render it a measure space, we denote $\nu \ll \mu$ if $\nu$ is absolutely continuous with respect to $\mu$. Hellinger integrals are defined below:

**Definition C.5** (Hellinger integrals (Hellinger, 1909)). Let $\nu, \mu$ be two probability measures on $(\Omega, \mathcal{F})$ and satisfy $\nu \ll \mu$, and $\varphi_\alpha : \mathbb{R}^+ \to \mathbb{R}$ be defined as $\varphi_\alpha(x) = x^\alpha$. Then the Hellinger integral of order $\alpha$ is given by

$$H_\alpha(\nu \| \mu) = \int \left( \frac{d\nu}{d\mu} \right)^\alpha \mathrm{d}\mu.$$

It can be seen as a $\phi$-Divergence with a specific parametrised choice of $\phi$ (Liese & Vajda, 2006). For $\alpha > 1$, the Hellinger integral measures the divergence between two probability distributions (Liese & Vajda, 2006). There holds $H_\alpha(\nu \| \mu) \in [1, +\infty), \alpha > 1$, and it equals to 1 if the two measures coincide (Shiryaev, 2016). Given such a divergence measure, we now provide the proof.

*Proof.* From Section C.1, we know that

$$
\begin{aligned}
TE(z) = L_P(z^*) - L_P(z) &\leq L_P(z^*) - L_P(z) + (L_E(z) - L_E(z^*)) \\
&= (L_P(z^*) - L_E(z^*)) + (L_E(z) - L_P(z)) \\
&\leq \sup_{x \in \mathcal{B}_\epsilon(x)} (L_P(z) - L_E(z)) + \sup_{x \in \mathcal{B}_\epsilon(x)} (L_E(z) - L_P(z)) \\
&\leq \sup_{z \in \mathcal{Z}} (L_P(z) - L_E(z)) + \sup_{z \in \mathcal{Z}} (L_E(z) - L_P(z)).
\end{aligned}
$$

Let $\theta^N = (\theta_1, \ldots, \theta_N), \theta'^N = (\theta'_1, \ldots, \theta'_N)$ that satisfy $\theta^N, \theta'^N \sim \mathcal{P}_{\Theta^N}$, and the $m$-th member is different, i.e., $\theta'_m \neq \theta_m$. We define

$$L_{E'}(z) = \frac{1}{N} \sum_{i=1}^{N} \ell(f(\theta'_i; x), y),$$

and

$$
\begin{aligned}
\Phi_1(E) &= \sup_{z \in \mathcal{Z}} \{ L_P(z) - L_E(z) \}, \\
\Phi_1(E') &= \sup_{z \in \mathcal{Z}} \{ L_P(z) - L_{E'}(z) \}.
\end{aligned}
$$

We have

$$\Phi_1(E) - \Phi_1(E') = \sup_{z \in \mathcal{Z}} \{ L_P(z) - L_E(z) \} - \sup_{z \in \mathcal{Z}} \{ L_P(z) - L_{E'}(z) \}$$

$$\leq \sup_{z \in \mathcal{Z}} \{L_P(z) - L_E(z) - (L_P(z) - L_{E'}(z))\}$$

$$= \sup_{z \in \mathcal{Z}} \{L_{E'}(z) - L_E(z)\}$$

$$= \frac{1}{N} \sup_{z \in \mathcal{Z}} \left[ \sum_{i=1}^{N} \ell(f(\theta_i'; x), y) - \sum_{i=1}^{N} \ell(f(\theta_i; x), y) \right].$$

By assuming that loss function $\ell$ is bounded by $\beta$, we have

$$|\Phi_1(E) - \Phi_1(E')| \leq \frac{\beta}{N}.$$

According to Theorem 1 in Esposito & Mondelli (2024), for all $\delta \in (0, 1)$ and $\alpha > 1$, with probability at least $1 - \frac{1}{4}\delta$, we have

$$\Phi_1(E) \leq \mathbb{E}_{\theta^N}[\Phi_1(E)] + \sqrt{\frac{\alpha \beta^2}{2(\alpha - 1)N} \ln \frac{2^{\frac{\alpha-1}{\alpha}} H_\alpha^{\frac{1}{\alpha}} \left( \mathcal{P}_{\Theta^N} \| \mathcal{P}_{\otimes_{i=1}^N \Theta} \right)}{\frac{1}{4}\delta}}. \tag{24}$$

Denote $f(\theta_i; x)$ as $f_i(x)$ and $f(\theta_i'; x)$ as $f_i'(x)$. Then we estimate the upper bound of $\mathbb{E}_{\theta^N \sim \mathcal{P}_{\Theta^N}}[\Phi_1(E)]$ as follows:

$$\mathbb{E}_{\theta^N}[\Phi_1(E)] = \mathbb{E}_{\theta^N} \left[ \sup_{z \in \mathcal{Z}} (L_P(z) - L_E(z)) \right]$$

$$= \mathbb{E}_{\theta^N} \left[ \sup_{z \in \mathcal{Z}} \mathbb{E}_{(\theta_1', \cdots, \theta_N') \sim \mathcal{P}_{\Theta^N}'} (L_{E'}(z) - L_E(z)) \right]$$

$$\leq \mathbb{E}_{\theta^N, \theta'^N} \left[ \sup_{z \in \mathcal{Z}} (L_{E'}(z) - L_E(z)) \right] \qquad \text{(Jensen inequality)}$$

$$= \mathbb{E}_{\theta^N, \theta'^N} \left\{ \sup_{z \in \mathcal{Z}} \frac{1}{N} \left[ \sum_{i=1}^{N} \ell(f(\theta_i'; x), y) - \sum_{i=1}^{N} \ell(f(\theta_i; x), y) \right] \right\}$$

$$= \mathbb{E}_{\boldsymbol{\sigma}} \mathbb{E}_{\theta^N, \theta'^N} \left\{ \sup_{z \in \mathcal{Z}} \frac{1}{N} \left[ \sum_{i=1}^{N} \sigma_i \left[ \ell(f_i'(x), y) - \ell(f_i(x), y) \right] \right] \right\}$$

$$\leq \mathbb{E}_{\boldsymbol{\sigma}} \mathbb{E}_{\theta'^N} \left\{ \sup_{z \in \mathcal{Z}} \frac{1}{N} \left[ \sum_{i=1}^{N} \sigma_i \ell(f_i'(x), y) \right] \right\} + \mathbb{E}_{\boldsymbol{\sigma}} \mathbb{E}_{\theta^N} \left\{ \sup_{z \in \mathcal{Z}} \frac{1}{N} \left[ \sum_{i=1}^{N} \sigma_i \ell(f_i(x), y) \right] \right\}$$

$$= 2 \cdot \mathbb{E}_{\boldsymbol{\sigma}} \mathbb{E}_{\theta^N} \left\{ \sup_{z \in \mathcal{Z}} \frac{1}{N} \sum_{i=1}^{N} \sigma_i \ell(f_i(x), y) \right\}$$

$$= 2\mathbb{E}_{\theta^N} \left[ \mathcal{R}_N(\mathcal{F}) \right]. \tag{25}$$

Since changing one element in $\theta^N$ changes $\mathcal{R}_N(\mathcal{F})$ by at most $\frac{\beta}{N}$, we again apply Theorem 1 in Esposito & Mondelli (2024) and obtain that for all $\delta \in (0, 1)$, with probability at least $1 - \frac{1}{4}\delta$, we have

$$\mathbb{E}_{\theta^N} \left[ \mathcal{R}_N(\mathcal{F}) \right] \leq \mathcal{R}_N(\mathcal{F}) + \sqrt{\frac{\alpha \beta^2}{2(\alpha - 1)N} \ln \frac{2^{\frac{\alpha-1}{\alpha}} H_\alpha^{\frac{1}{\alpha}} \left( \mathcal{P}_{\Theta^N} \| \mathcal{P}_{\otimes_{i=1}^N \Theta} \right)}{\frac{1}{4}\delta}}. \tag{26}$$

Likewise, if we define

$$\Phi_2(E) = \sup_{z \in \mathcal{Z}} \{L_E(z) - L_P(z)\},$$

$$\Phi_2(E') = \sup_{z \in \mathcal{Z}} \{L_{E'}(z) - L_P(z)\},$$

then we have

$$
\begin{aligned}
\Phi_2(E) - \Phi_2(E') &= \sup_{z \in \mathcal{Z}} \{L_E(z) - L_P(z)\} - \sup_{z \in \mathcal{Z}} \{L_{E'}(z) - L_P(z)\} \\
&\leq \sup_{z \in \mathcal{Z}} \{L_E(z) - L_P(z) - (L_{E'}(z) - L_P(z))\} \\
&= \sup_{z \in \mathcal{Z}} \{L_E(z) - L_{E'}(z)\} \\
&= \frac{1}{N} \sup_{z \in \mathcal{Z}} \left[ \sum_{i=1}^{N} \ell(f(\theta_i; x), y) - \sum_{i=1}^{N} \ell(f(\theta_i'; x), y) \right].
\end{aligned}
$$

According to the assumption that loss function $\ell$ is bounded by $\beta$, we have

$$
|\Phi_2(E) - \Phi_2(E')| \leq \frac{\beta}{N}.
$$

According to Theorem 1 in Esposito & Mondelli (2024), for all $\delta \in (0, 1)$ and $\alpha > 1$, with probability at least $1 - \frac{1}{4}\delta$, we have

$$
\Phi_2(E) \leq \mathbb{E}_{\theta^N}[\Phi_2(E)] + \sqrt{\frac{\alpha\beta^2}{2(\alpha-1)N} \ln \frac{2^{\frac{\alpha-1}{\alpha}} H_{\alpha}^{\frac{1}{\alpha}}\left(\mathcal{P}_{\Theta^N} \| \mathcal{P}_{\bigotimes_{i=1}^{N} \Theta_i}\right)}{\frac{1}{4}\delta}}. \tag{27}
$$

We estimate the upper bound of $\mathbb{E}_{\theta^N}[\Phi_2(E)]$ as follows:

$$
\begin{aligned}
\mathbb{E}_{\theta^N}[\Phi_2(E)] &= \mathbb{E}_{\theta^N}\left[\sup_{z \in \mathcal{Z}}(L_E(z) - L_P(z))\right] \\
&= \mathbb{E}_{\theta^N}\left[\sup_{z \in \mathcal{Z}} \mathbb{E}_{(\theta_1', \cdots, \theta_N') \sim \mathcal{P}_{\Theta^N}'}(L_E(z) - L_{E'}(z))\right] \\
&\leq \mathbb{E}_{\theta^N, \theta'^N}\left[\sup_{z \in \mathcal{Z}}(L_E(z) - L_{E'}(z))\right] \qquad \text{(Jensen inequality)} \\
&= \mathbb{E}_{\theta^N, \theta'^N}\left\{\sup_{z \in \mathcal{Z}} \frac{1}{N}\left[\sum_{i=1}^{N} \ell(f(\theta_i; x), y) - \sum_{i=1}^{N} \ell(f(\theta_i'; x), y)\right]\right\} \\
&= \mathbb{E}_{\boldsymbol{\sigma}} \mathbb{E}_{\theta^N, \theta'^N}\left\{\sup_{z \in \mathcal{Z}} \frac{1}{N}\left[\sum_{i=1}^{N} \sigma_i\left[\ell(f_i(x), y) - \ell(f_i'(x), y)\right]\right]\right\} \\
&\leq \mathbb{E}_{\boldsymbol{\sigma}} \mathbb{E}_{\theta'^N}\left\{\sup_{z \in \mathcal{Z}} \frac{1}{N}\left[\sum_{i=1}^{N} \sigma_i\ell(f_i'(x), y)\right]\right\} + \mathbb{E}_{\boldsymbol{\sigma}} \mathbb{E}_{\theta^N}\left\{\sup_{z \in \mathcal{Z}} \frac{1}{N}\left[\sum_{i=1}^{N} \sigma_i\ell(f_i(x), y)\right]\right\} \\
&= 2 \cdot \mathbb{E}_{\boldsymbol{\sigma}} \mathbb{E}_{\theta^N}\left\{\sup_{z \in \mathcal{Z}} \frac{1}{N} \sum_{i=1}^{N} \sigma_i\ell(f_i(x), y)\right\} \\
&= 2\mathbb{E}_{\theta^N}[\mathcal{R}_N(\mathcal{F})]. \tag{28}
\end{aligned}
$$

Likewise, we again apply Theorem 1 in Esposito & Mondelli (2024) and obtain that for all $\delta \in (0, 1)$, with probability at least $1 - \frac{1}{4}\delta$, we have

$$
\mathbb{E}_{\theta^N}[\mathcal{R}_N(\mathcal{F})] \leq \mathcal{R}_N(\mathcal{F}) + \sqrt{\frac{\alpha\beta^2}{2(\alpha-1)N} \ln \frac{2^{\frac{\alpha-1}{\alpha}} H_{\alpha}^{\frac{1}{\alpha}}\left(\mathcal{P}_{\Theta^N} \| \mathcal{P}_{\bigotimes_{i=1}^{N} \Theta}\right)}{\frac{1}{4}\delta}}. \tag{29}
$$

Therefore, combining Eq. (24), Eq. (25), Eq. (26), Eq. (27), Eq. (28) and Eq. (29) with union bound, we obtain that, with probability at least $1 - \delta$, there holds

$$TE(z,\epsilon) = \Phi_1(E) + \Phi_2(E) \le 4\mathcal{R}_N(\mathcal{F}) + \sqrt{\frac{18\alpha\beta^2}{(\alpha-1)N} \ln \frac{2^{2+\frac{\alpha-1}{\alpha}} H_\alpha^{\frac{1}{\alpha}} \left(\mathcal{P}_{X^n} \| \mathcal{P}_{\otimes_{i=1}^n X_i}\right)}{\delta}}.$$

The proof is complete.

$\square$

## C.5   Extension of Theorem 4.3

We consider $N$ surrogate classifiers $f_1, \cdots, f_N$ trained to generate adversarial examples. Let $D$ be the distribution over the surrogate models (for instance, the distribution of all the low-risk models), and $f_i \in D, i \in [N]$. The low-risk claim is in line with Lemma 5 in (Yang et al., 2021), which assumes that the risk of surrogate model and target model is low (have risk at most $\epsilon$). Therefore, the surrogate model and target model can be seen as drawing from the same distribution (such as a distribution of all the low-risk models). For a data point $z = (x, y) \in \mathcal{Z}$ and $N$ classifiers for model ensemble attack, define the population risk $L_P(z)$ and the empirical risk $L_D(z)$ as

$$L_P(z) = \mathbb{E}_{f \sim D}[\ell(f(x), y)].$$

$$L_D(z) = \frac{1}{N} \sum_{i \in [N], f_i \in D} \ell(f_i(x), y).$$

Now here is an extension of Theorem 4.3 based on the above definition.

**Theorem C.6** (Extension of Theorem 4.3). *Let $\mathcal{P}_{D^N}$ be the joint distribution of $f_1, \cdots, f_N$, and $\mathcal{P}_{\otimes_{i=1}^N D}$ be the joint measure induced by the product of the marginals. If the loss function $\ell$ is bounded by $\beta \in R_+$ and $\mathcal{P}_{D^N} \ll \mathcal{P}_{\otimes_{i=1}^N D}$ for any function $f_i$, then for $\alpha > 1$ and $\gamma = \frac{\alpha}{\alpha-1}$, with probability at least $1 - \delta$, there holds*

$$TE(z,\epsilon) \le 4\mathcal{R}_N(\mathcal{Z}) + \sqrt{\frac{18\gamma\beta^2}{N} \ln \frac{2^{2+\frac{1}{\gamma}} H_\alpha^{\frac{1}{\alpha}} \left(\mathcal{P}_{D^N} \| \mathcal{P}_{\otimes_{i=1}^N D}\right)}{\delta}}. \tag{30}$$

The proof is almost the same as Appendix C.4, but the definition of distribution is different. The first term answers the question that more surrogate models and smaller complexity will lead to a smaller $\mathcal{R}_N(\mathcal{Z})$ and contributes to a tighter bound of $TE(z, \epsilon)$. The second term motivates us that if we reduce the interdependency among the ensemble components, then the upper bound of $TE(z, \epsilon)$ will be tighter. Recall that $H_\alpha(\mathcal{P}_{D^N} \| \mathcal{P}_{\otimes_{i=1}^N D})$ quantifies the divergence between the joint distribution $\mathcal{P}_{D^N}$ and product of marginals $\mathcal{P}_{\otimes_{i=1}^N D}$. The joint distribution captures dependencies while the product of marginals does not. So the divergence between them measures the degree of dependency among the $N$ classifiers $f_1, \cdots, f_N$. As a result, improving the diversity of $f_1, \cdots, f_N$ and reduce the interdependence among them is beneficial to adversarial transferability.

## C.6   Further Explanation of the Hellinger Integral Term

We provide two examples of the Hellinger integral term in Theorem 4.3.

*Example* 1 (Independent case). Suppose that the $N$ surrogate models are independent. In this case, the hellinger integral achieves its minimum 1. Therefore, let $\alpha = 2$ and Theorem 4.3 becomes

$$TE(z,\epsilon) \le 4\mathcal{R}_N(\mathcal{Z}) + \sqrt{\frac{36\beta^2}{N} \ln \frac{4\sqrt{2}}{\delta}}.$$

This theoretical result is similar to the generalization error bound in the literature on statistical learning theory (Bartlett & Mendelson, 2002) with different constant coefficients. The difference arises because (Bartlett & Mendelson, 2002) applies the concentration inequality once, but our proof applies it several times.

*Example* 2 (Dependent case). For a more general case, the $N$ surrogate models are interdependent to each other. While it is hard to model the behavior of each model and the whole parameter space, we simplify the problem to make it clear

to understand the hellinger integral $H_\alpha(\mathcal{P}_{\Theta^N} \| \mathcal{P}_{\otimes_{i=1}^N \Theta})$. In particular, let $P = \mathcal{P}_{\Theta^N}$ and $Q = \mathcal{P}_{\otimes_{i=1}^N \Theta}$. We consider the model parameters for a given precision so that $P$ and $Q$ are discrete distributions. Firstly, Equation (8) from (Esposito & Mondelli, 2024) tells us that $H_\alpha(P\|Q) = e^{(\alpha-1)D_\alpha(P,Q)}$, where $D_\alpha(P,Q)$ is the Rényi divergence. Secondly, let $\beta_1 = \min_{a \in \mathcal{A}} \frac{Q(a)}{P(a)}$ be defined in Equation (8) from (Sason & Verdú, 2015), i.e., the minimum of the ratio of the probability density function of distributions $Q$ and $P$. Now we approximate $\beta_1$. Consider there are $t$ parameter configurations for each model. For simplicity, we assume that part of the models ($f(N)$ models) play a key role in adversarial transferability, and the other $N - f(N)$ models are random sampled from these $f(N)$ models.

- For the product of marginal distribution $Q$, the parameters from each model are random. Consider the case of uniform distribution, where every parameter in the $N$ models share the same probability, i.e., $Q(a) = \frac{1}{t^N}$.

- For the joint distribution $P$, we also consider the case of uniform distribution, where $f(N)$ models are fixed and $N - f(N)$ models are randomly sampled, i.e., $P(a) = \frac{1}{t^{N-f(N)}}$.

Therefore, $\beta_1 = \frac{Q(a)}{P(a)} = t^{-f(N)}$, which is less than 1. Substitute the above into Theorem 3 from (Sason & Verdú, 2015), we have

$$H_\alpha(P\|Q) \le 1 + \frac{\mathrm{D_{TV}}(P\|Q) \cdot (\beta_1^{-1} - 1)}{1 - \beta_1} \le 1 + \frac{(\beta_1^{-1} - 1)}{1 - \beta_1} \le \beta_1^{-1} = t^{f(N)}.$$

Let $\alpha = 2$ and substitute the above into Theorem 4.3 in our paper, we have

$$TE(z, \epsilon) \le 4\mathcal{R}_N(\mathcal{Z}) + \sqrt{18\beta^2 \ln t \cdot \frac{f(N)}{N} + 36\beta^2 \ln \frac{4\sqrt{2}}{\delta} \cdot \frac{1}{N}}$$

Here are several cases:

1. $f(N) = \mathcal{O}(N^s)$, where $s \in (0, 1)$,

2. $f(N) = \mathcal{O}(\ln N)$,

3. $f(N) = sN$, where $s \in (0, 1)$.

For Cases 1 and 2, the above term asymptotically converges to zero as $N$ becomes large. Notably, the true Hellinger term may be smaller than our derived upper bound above. Quantifying the core subset of models $f(N)$ that dominate the performance of the ensemble attack presents a theoretically profound and practically significant research direction. This problem is particularly well-suited for future exploration, as it could fundamentally advance our understanding of transferable adversarial model ensemble attacks.

## C.7 Information-theoretic Analysis

This section follows the multi-classification setting in Appendix C.3. Note that while we use a different theoretical framework comparing to Theorem 4.3, the conclusion is consistent with it.

Firstly, we define the KL divergence, mutual information and TV distance.

**Definition C.7** (Kullback-Leibler Divergence). Given two probability distributions $P$ and $Q$, the Kullback-Leibler (KL) divergence between $P$ and $Q$ is

$$\mathrm{D_{KL}}(P\|Q) = \int_{x \in \mathcal{X}} P(x) \log \frac{P(x)}{Q(x)} dx.$$

We know that $\mathrm{D_{KL}}(P\|Q) \in [0, +\infty]$, and $\mathrm{D_{KL}}(P\|Q) = 0$ if and only if $P = Q$.

**Definition C.8** (Mutual Information). For continuous random variables $X$ and $Y$ with joint probability density function $p(x, y)$ and marginal probability density functions $p(x)$ and $p(y)$, the mutual information is defined as:

$$I(X; Y) = \iint p(x, y) \log \frac{p(x, y)}{p(x)p(y)} dx dy.$$

We know that $I(X;Y) \in [0, +\infty]$, and $I(X;Y) = 0$ if and only if $X$ and $Y$ are independent to each other.

**Definition C.9** (Total Variation Distance)**.** Given two probability distributions $P$ and $Q$, the Total Variation (TV) distance between $P$ and $Q$ is

$$D_{\mathrm{TV}}(P\|Q) = \frac{1}{2} \int_{x \in \mathcal{X}} |P(x) - Q(x)| \, dx.$$

We know that $D_{\mathrm{TV}}(P\|Q) \in [0, 1]$. Also, $D_{\mathrm{TV}}(P\|Q) = 0$ if and only if $P$ and $Q$ coincides, and $D_{\mathrm{TV}}(P\|Q) = 1$ if and only if $P$ and $Q$ are disjoint.

Here we provide further analysis from the perspective of information (Shwartz-Ziv & Tishby, 2017; Xu & Raginsky, 2017).

**Theorem C.10.** *Given $N$ surrogate models $\boldsymbol{\theta}^N \sim \mathcal{P}_{\Theta^N}$ as the ensemble components. Let $\overline{\boldsymbol{\theta}}^N = (\overline{\boldsymbol{\theta}}_1, \ldots, \overline{\boldsymbol{\theta}}_N) \sim \mathcal{P}_{\Theta^N}$ be the target models, which is an independent copy of $\boldsymbol{\theta}^N$. Assume the loss function $\ell$ is bounded by $\beta \in \mathbb{R}_+$ and $\mathcal{P}_{\Theta^N}$ is absolutely continuous with respect to $\mathcal{P}_{\otimes_{i=1}^N \Theta}$. For $\alpha > 1$ and adversarial example $\mathbf{z} = (\mathbf{x}, \mathbf{y}) \sim \mathcal{P}_{\mathcal{Z}}$, Let $\Delta_N(\boldsymbol{\theta}, \mathbf{z}) = L_P(\mathbf{z}) - L_E(\mathbf{z})$. Then there holds*

$$\left| \mathbb{E}_{\mathbf{z}, \boldsymbol{\theta}^N \sim \mathcal{P}_{\mathcal{Z}, \Theta^N}} \Delta_N(\boldsymbol{\theta}, \mathbf{z}) \right| \leq 2\beta \cdot D_{\mathrm{TV}} \left( \mathcal{P}_{\Theta^N} \| \mathcal{P}_{\otimes_{i=1}^N \Theta} \right) +$$

$$\sqrt{\frac{\alpha \beta^2}{2(\alpha - 1)N} \left( I\left( \boldsymbol{\theta}^N; z \right) + \frac{1}{\alpha} \log H_\alpha \left( \mathcal{P}_{\Theta^N} \| \mathcal{P}_{\otimes_{i=1}^N \Theta} \right) \right)},$$

*where $D_{\mathrm{TV}}(\cdot\|\cdot)$, $I(\cdot\|\cdot)$ and $H_\alpha(\cdot\|\cdot)$ denotes TV distance, mutual information and Hellinger integrals, respectively.*

In Theorem C.10: $\Delta_N(\boldsymbol{\theta}, \mathbf{z})$ quantifies how effectively the surrogate models represent all possible target models. Taking the expectation of $\Delta_N(\boldsymbol{\theta}, \mathbf{z})$ over $\mathbf{z}$ and $\boldsymbol{\theta}^N$ accounts for the inherent randomness in both adversarial examples and surrogate models. The mutual information $I\left( \boldsymbol{\theta}^N; \mathbf{z} \right)$ quantifies how much information about the surrogate models is retained in the adversarial example. Intuitively, higher mutual information indicates that the adversarial example is overly tailored to the surrogate models, capturing specific features of these models. This overfitting reduces its ability to generalize and transfer effectively to other target models. By controlling the complexity of the surrogate models, the specific information captured by the adversarial example can be limited, encouraging it to rely on broader, more transferable patterns rather than model-specific details. This reduction in overfitting enhances the adversarial example's transferability to diverse target models. The TV distance $D_{\mathrm{TV}} \left( \mathcal{P}_{\Theta^N} \| \mathcal{P}_{\otimes_{i=1}^N \Theta} \right)$ and the Hellinger integral $H_\alpha \left( \mathcal{P}_{\Theta^N} \| \mathcal{P}_{\otimes_{i=1}^N \Theta} \right)$ capture the interdependence among the surrogate models.

Theorem C.10 reveals that the following strategies contribute to a tighter bound: 1) Increasing the number of surrogate models, i.e., increasing $N$; 2) Reducing the model complexity of surrogate models, i.e., reducing $I\left( \boldsymbol{\theta}^N; \mathbf{z} \right)$; 3) Making the surrogate models more diverse, i.e., reducing $D_{\mathrm{TV}} \left( \mathcal{P}_{\Theta^N} \| \mathcal{P}_{\otimes_{i=1}^N \Theta} \right)$ and $H_\alpha \left( \mathcal{P}_{\Theta^N} \| \mathcal{P}_{\otimes_{i=1}^N \Theta} \right)$. A tighter bound ensures that an adversarial example maximizing the loss function on the surrogate models will also lead to a high loss on the target models, thereby enhancing transferability.

*Proof.* According to Donsker and Varadhan's variational formula, for any $\lambda \in \mathbb{R}$, there holds:

$$D_{\mathrm{KL}}(\mathcal{P}_{\mathcal{Z}, \Theta^N} \| \mathcal{P}_{\mathcal{Z}} \otimes \mathcal{P}_{\Theta^N}) \geq \lambda \mathbb{E}_{\mathbf{z}, \boldsymbol{\theta}^N \sim \mathcal{P}_{\mathcal{Z}, \Theta^N}} \Delta_N(\boldsymbol{\theta}, \mathbf{z}) - \log \mathbb{E}_{\mathbf{z} \sim \mathcal{P}_{\mathcal{Z}}} \mathbb{E}_{\boldsymbol{\theta}^N \sim \mathcal{P}_{\Theta^N}} \left[ e^{\lambda \Delta_N(\boldsymbol{\theta}, \mathbf{z})} \right]. \tag{31}$$

Fix $z \in \mathcal{Z}$,

$$\mathbb{E}_{\boldsymbol{\theta}^N \sim \mathcal{P}_{\Theta^N}} \left[ e^{\lambda \Delta_N(\boldsymbol{\theta}, \mathbf{z})} \right] = \int e^{\lambda \Delta_N(\boldsymbol{\theta}, \mathbf{z})} d\mathcal{P}_{\Theta^N}$$

$$= \int e^{\lambda \Delta_N(\boldsymbol{\theta}, \mathbf{z})} \frac{d\mathcal{P}_{\Theta^N}}{d\mathcal{P}_{\otimes_{i=1}^N \Theta}} d\mathcal{P}_{\otimes_{i=1}^N \Theta}$$

$$\leq \left( \int e^{\frac{\alpha}{\alpha - 1} \lambda \Delta_N(\boldsymbol{\theta}, \mathbf{z})} d\mathcal{P}_{\otimes_{i=1}^N \Theta} \right)^{\frac{\alpha - 1}{\alpha}} \left( \int \left( \frac{d\mathcal{P}_{\Theta^N}}{d\mathcal{P}_{\otimes_{i=1}^N \Theta}} \right)^\alpha d\mathcal{P}_{\otimes_{i=1}^N \Theta} \right)^{\frac{1}{\alpha}}$$

$$= \left( \int e^{\frac{\alpha}{\alpha-1}\lambda\Delta_N(\boldsymbol{\theta},\mathbf{z})} d\mathcal{P}_{\otimes_{i=1}^N \Theta} \right)^{\frac{\alpha-1}{\alpha}} H_\alpha^{\frac{1}{\alpha}}(\mathcal{P}_{\Theta^N} \| \mathcal{P}_{\otimes_{i=1}^N \Theta}). \tag{32}$$

The third line uses Hölder's inequality, while the last line follows Definition C.5. Now we deal with the first term. Denote

$$\Delta_1 = \mathbb{E}_{\overline{\boldsymbol{\theta}}^N \sim \mathcal{P}_{\Theta^N}} \left[ \frac{1}{N} \sum_{i=1}^N \ell(f(\overline{\boldsymbol{\theta}}_i; \mathbf{x}), \mathbf{y}) \right] - \mathbb{E}_{\overline{\boldsymbol{\theta}}^N \sim \mathcal{P}_{\otimes_{i=1}^N \Theta}} \left[ \frac{1}{N} \sum_{i=1}^N \ell(f(\overline{\boldsymbol{\theta}}_i; \mathbf{x}), \mathbf{y}) \right],$$

$$\Delta_2 = \mathbb{E}_{\overline{\boldsymbol{\theta}}^N \sim \mathcal{P}_{\otimes_{i=1}^N \Theta}} \left[ \frac{1}{N} \sum_{i=1}^N \ell(f(\overline{\boldsymbol{\theta}}_i; \mathbf{x}), \mathbf{y}) \right] - \frac{1}{N} \sum_{i=1}^N \ell(f(\boldsymbol{\theta}_i; \mathbf{x}), \mathbf{y}).$$

Notice that

$$|\Delta_1| = \left| \iint \cdots \int \left[ \frac{1}{N} \sum_{i=1}^N \ell(f(\overline{\boldsymbol{\theta}}_i; \mathbf{x}), \mathbf{y}) \right] \left[ \mathcal{P}_{\Theta^N}(\overline{\boldsymbol{\theta}}_1, \cdots, \overline{\boldsymbol{\theta}}_N) - \mathcal{P}_{\otimes_{i=1}^N \Theta}(\overline{\boldsymbol{\theta}}_1, \cdots, \overline{\boldsymbol{\theta}}_N) \right] d\overline{\boldsymbol{\theta}}_1 \cdots d\overline{\boldsymbol{\theta}}_N \right|$$

$$\leq \beta \iint \cdots \int \left| \mathcal{P}_{\Theta^N}(\overline{\boldsymbol{\theta}}_1, \cdots, \overline{\boldsymbol{\theta}}_N) - \mathcal{P}_{\otimes_{i=1}^N \Theta}(\overline{\boldsymbol{\theta}}_1, \cdots, \overline{\boldsymbol{\theta}}_N) \right| d\overline{\boldsymbol{\theta}}_1 \cdots d\overline{\boldsymbol{\theta}}_N$$

$$= \beta \int \left| \mathcal{P}_{\Theta^N}\left(\overline{\boldsymbol{\theta}}^N\right) - \mathcal{P}_{\otimes_{i=1}^N \Theta}\left(\overline{\boldsymbol{\theta}}^N\right) \right| d\overline{\boldsymbol{\theta}}^N$$

$$\leq 2\beta \cdot \mathrm{D_{TV}}\left(\mathcal{P}_{\Theta^N} \| \mathcal{P}_{\otimes_{i=1}^N \Theta}\right). \tag{33}$$

Also,

$$\int \left( e^{\frac{\alpha}{\alpha-1}\lambda\Delta_2} \right) d\mathcal{P}_{\otimes_{i=1}^N \Theta} = \mathbb{E}_{\boldsymbol{\theta}^N \sim \mathcal{P}_{\otimes_{i=1}^N \Theta}} \left[ e^{\frac{\alpha}{\alpha-1}\lambda\Delta_2} \right]$$

$$= \prod_{i=1}^N \mathbb{E}_{\boldsymbol{\theta}_i \sim \mathcal{P}_\Theta} \left[ \exp\left( \frac{\alpha\lambda}{\alpha-1} \left( \mathbb{E}_{\overline{\boldsymbol{\theta}}_i \sim \mathcal{P}_\Theta} \left[ \frac{1}{N}\ell(f(\overline{\boldsymbol{\theta}}_i; \mathbf{x}), \mathbf{y}) \right] - \frac{1}{N}\ell(f(\boldsymbol{\theta}_i; \mathbf{x}), \mathbf{y}) \right) \right) \right]$$

$$\leq \prod_{i=1}^N \exp\left( \frac{\alpha^2}{8(\alpha-1)^2 N^2} \lambda^2 \beta^2 \right).$$

$$\leq \exp\left( \frac{\alpha^2}{8(\alpha-1)^2 N} \lambda^2 \beta^2 \right). \tag{34}$$

The third line is due to Hoeffding's Lemma (using it for each $\boldsymbol{\theta}_i$). Therefore, recall the fact that $\Delta_N(\boldsymbol{\theta}, \mathbf{z}) = \Delta_1 + \Delta_2$, we have

$$\int e^{\frac{\alpha}{\alpha-1}\lambda\Delta_N(\boldsymbol{\theta},\mathbf{z})} d\mathcal{P}_{\otimes_{i=1}^N \Theta} = \int \left( e^{\frac{\alpha}{\alpha-1}\lambda\Delta_1} \cdot e^{\frac{\alpha}{\alpha-1}\lambda\Delta_2} \right) d\mathcal{P}_{\otimes_{i=1}^N \Theta}$$

$$\leq \exp\left( \frac{2\lambda\alpha\beta}{\alpha-1} \mathrm{D_{TV}}\left(\mathcal{P}_{\Theta^N} \| \mathcal{P}_{\otimes_{i=1}^N \Theta}\right) \right) \int e^{\frac{\alpha}{\alpha-1}\lambda\Delta_2} d\mathcal{P}_{\otimes_{i=1}^N \Theta} \qquad \text{(Using (33))}$$

$$\leq \exp\left( \frac{2\lambda\alpha\beta}{\alpha-1} \mathrm{D_{TV}}\left(\mathcal{P}_{\Theta^N} \| \mathcal{P}_{\otimes_{i=1}^N \Theta}\right) + \frac{\alpha^2}{8(\alpha-1)^2 N} \lambda^2 \beta^2 \right) \qquad \text{(Using (34))}$$

With the above results, we obtain the following:

$$\log \mathbb{E}_{\mathbf{z} \sim \mathcal{P}_\mathcal{Z}} \mathbb{E}_{\boldsymbol{\theta}^N \sim \mathcal{P}_{\Theta^N}} \left[ e^{\lambda\Delta_N(\boldsymbol{\theta},\mathbf{z})} \right] \leq 2\lambda\beta \cdot \mathrm{D_{TV}}\left(\mathcal{P}_{\Theta^N} \| \mathcal{P}_{\otimes_{i=1}^N \Theta}\right) + \frac{\alpha}{8(\alpha-1)N} \lambda^2 \beta^2 + \log H_\alpha^{\frac{1}{\alpha}}(\mathcal{P}_{\Theta^N} \| \mathcal{P}_{\otimes_{i=1}^N \Theta}).$$

Substitute the above into Eq. (31), we have

$$\frac{\alpha}{8(\alpha-1)N} \beta^2 \lambda^2 + \left( 2\beta \cdot \mathrm{D_{TV}}\left(\mathcal{P}_{\Theta^N} \| \mathcal{P}_{\otimes_{i=1}^N \Theta}\right) - \mathbb{E}_{\mathbf{z},\boldsymbol{\theta}^N \sim \mathcal{P}_{\mathcal{Z},\Theta^N}} \Delta_N(\boldsymbol{\theta}, \mathbf{z}) \right) \lambda +$$

$$\mathrm{D_{KL}}(\mathcal{P}_{\mathcal{Z},\Theta^N} \| \mathcal{P}_\mathcal{Z} \otimes \mathcal{P}_{\Theta^N}) + \log H_\alpha^{\frac{1}{\alpha}}(\mathcal{P}_{\Theta^N} \| \mathcal{P}_{\otimes_{i=1}^N \Theta}) \geq 0.$$

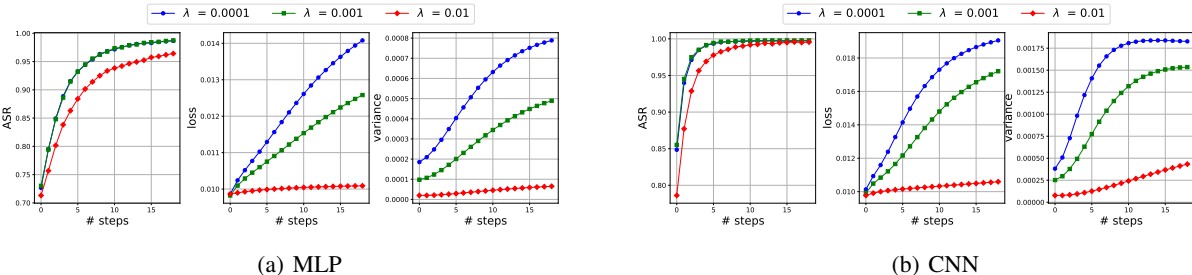

(a) MLP        (b) CNN

*Figure 6.* Evaluation of ensemble attacks with increasing the number of steps using MLPs and CNNs on the CIFAR-100 dataset.

Let the discriminant of the quadratic function with respect to $\lambda$ be less than or equal to $0$, leading to:

$$\left| 2\beta \cdot \mathrm{D}_{\mathrm{TV}} \left( \mathcal{P}_{\Theta^N} \| \mathcal{P}_{\bigotimes_{i=1}^N \Theta} \right) - \mathbb{E}_{\mathbf{z}, \boldsymbol{\theta}^N \sim \mathcal{P}_{\mathcal{Z}, \Theta^N}} \Delta_N(\boldsymbol{\theta}, \mathbf{z}) \right| \leq$$

$$\sqrt{ \frac{\alpha \beta^2}{2(\alpha-1)N} \left( \mathrm{D}_{\mathrm{KL}} \left( \mathcal{P}_{\mathcal{Z}, \Theta^N} \| \mathcal{P}_{\mathcal{Z}} \otimes \mathcal{P}_{\Theta^N} \right) + \frac{1}{\alpha} \log H_\alpha \left( \mathcal{P}_{\Theta^N} \| \mathcal{P}_{\bigotimes_{i=1}^N \Theta} \right) \right) }. \quad (35)$$

In other words,

$$\left| \mathbb{E}_{z, \boldsymbol{\theta}^N \sim \mathcal{P}_{\mathcal{Z}, \Theta^N}} \Delta_N(\boldsymbol{\theta}, z) \right| \leq 2\beta \cdot \mathrm{D}_{\mathrm{TV}} \left( \mathcal{P}_{\Theta^N} \| \mathcal{P}_{\bigotimes_{i=1}^N \Theta} \right) +$$

$$\sqrt{ \frac{\alpha \beta^2}{2(\alpha-1)N} \left( \mathrm{D}_{\mathrm{KL}} \left( \mathcal{P}_{\mathcal{Z}, \Theta^N} \| \mathcal{P}_{\mathcal{Z}} \otimes \mathcal{P}_{\Theta^N} \right) + \frac{1}{\alpha} \log H_\alpha \left( \mathcal{P}_{\Theta^N} \| \mathcal{P}_{\bigotimes_{i=1}^N \Theta} \right) \right) }.$$

Finally, substitute $I\left( \boldsymbol{\theta}^N; \mathbf{z} \right) = \mathrm{D}_{\mathrm{KL}} \left( \mathcal{P}_{\mathcal{Z}, \Theta^N} \| \mathcal{P}_{\mathcal{Z}} \otimes \mathcal{P}_{\Theta^N} \right)$ into above and we can get the desired result. $\qquad\square$

# D   Further Experiments

## D.1   Evaluation on CIFAR-100

Following the same setting in our experiments, we further validate the vulnerability-diversity decomposition on the CIFAR-100 (Krizhevsky et al., 2009) dataset. The results are shown in Figure 6. As the model becomes stronger (i.e., a smaller $\lambda$), the three metrics (ASR, loss and variance) increases, validating the soundness of vulnerability-diversity decomposition.

## D.2   Further Investigation into Model Complexity

We conduct a deeper investigation into the role of model complexity by applying a max norm constraint to the model parameters. Specifically, we constrain the $L_2$ norm of each weight vector to a predefined threshold, effectively limiting the model's capacity. Empirically, larger max norm values allow for more expressive feature representations but may increase the risk of overfitting. In contrast, smaller max norms encourage simpler models and reduce overfitting but may also lead to underfitting due to restricted representational power. The validation of this trade-off is illustrated in Table 1, which shows the classification accuracy across a range of max norm values for both MLP and CNN architectures with varying depths. Lower accuracy values indicate stronger adversarial attack performance.

The results reveal a consistent trend: as the max norm constraint is relaxed from a highly restrictive value (*e.g.*, 0.1) to a moderate level (*e.g.*, 5.0), the effectiveness of adversarial attacks improves and then declines. This observation indicates that overly tight constraints can impair model expressiveness, while moderately relaxed constraints can achieve a better trade-off between simplicity and capacity. These findings empirically support our theoretical claim that weight regularization, *e.g.*, via weight decay or norm bounds, directly influences model complexity and, consequently, adversarial transferability.

## D.3   Experiments on ImageNet

To further investigate how appropriately controlling the complexity of surrogate models contributes to effective adversarial attack algorithms—in line with our theoretical insights—we conduct additional experiments on ImageNet (Russakovsky

*Table 1.* Effect of varying max norm constraints on adversarial attack performance, measured by classification accuracy (%, lower is better). FC and CNN denote fully connected and convolutional networks with increasing layers.

| Max Norm | FC1 | FC2 | FC3 | CNN1 | CNN2 | CNN3 | Avg |
|---|---|---|---|---|---|---|---|
| 0.1 | 84.66 | 87.80 | 85.39 | 97.57 | 98.31 | 98.59 | 92.05 |
| 0.5 | 59.37 | 68.31 | 74.05 | 96.50 | 97.66 | 98.34 | 82.37 |
| 1.0 | 64.31 | 55.27 | 57.12 | 95.37 | 97.08 | 97.93 | 77.85 |
| 2.0 | 68.00 | 57.40 | 57.86 | 95.41 | 97.04 | 97.87 | 78.93 |
| 4.0 | 68.19 | 57.94 | 58.12 | 95.53 | 97.00 | 97.85 | 79.11 |
| 5.0 | 69.68 | 59.40 | 59.26 | 97.48 | 98.02 | 98.87 | 80.45 |

et al., 2015). For model ensemble attacks, we fine-tune several surrogate models, including VGG16 (Simonyan & Zisserman, 2014), Inception-V3 (Szegedy et al., 2016), and Visformer (Chen et al., 2021), using a sparse Softmax cross-entropy loss (Martins & Astudillo, 2016). This modification encourages sparsity in the model's output distribution. As shown in Table 2, this approach leads to a reduction in model complexity, as indicated by the decreased $L_2$ norm of the weights.

*Table 2.* Comparison of model complexity between original and sparse Softmax loss variants on different backbones. Lower values indicate reduced $L_2$ norm of weights.

| | VGG16 | Visformer | InceptionV3 |
|---|---|---|---|
| Original | 37.37 | 25.94 | 49.24 |
| Sparse Softmax Loss | 33.12 | 20.60 | 48.53 |

We then leverage these sparsified models for ensemble attacks by applying MI-FGSM (Dong et al., 2018), SVRE (Xiong et al., 2022), and SIA (Wang et al., 2023a) to both the original and sparsified versions, resulting in MI-FGSM-S, SVRE-S, and SIA-S, respectively. The transferability of these attacks is evaluated on a range of target models, and the results are presented in Table 3. As observed, these sparsified variants consistently outperform their standard counterparts in most cases, validating the advantage of model complexity control for enhancing adversarial transferability. This improvement holds across both CNN-based and transformer-based architectures. Beyond this example, our findings may inspire the design of stronger adversarial attack strategies that systematically exploit sparsity and simplicity in surrogate modeling.

*Table 3.* Transferability results of different attack methods across various target models. Bold entries indicate improved or top-performing variants.

| | ResNet50 | VGG16 | MobileNetV2 | InceptionV3 | ViT-B16 | PiT-B | Visformer | Swin-T |
|---|---|---|---|---|---|---|---|---|
| MI-FGSM | 66.0 | **99.9** | 76.8 | 97.5 | 37.3 | 53.8 | 88.9 | 66.7 |
| **MI-FGSM-S** | **68.9** | 99.7 | **79.2** | **99.1** | **39.0** | **54.5** | **90.6** | **68.1** |
| SVRE | 65.2 | **99.9** | 79.0 | 98.6 | 32.4 | 49.2 | 90.3 | 64.3 |
| **SVRE-S** | **66.9** | **99.9** | **81.2** | **98.9** | **34.2** | **51.3** | **93.0** | **65.9** |
| SIA | 97.2 | **100.0** | **98.4** | **99.7** | 75.9 | 91.9 | 90.0 | 96.1 |
| **SIA-S** | **98.1** | **100.0** | 98.2 | 99.6 | **79.2** | **93.2** | **99.5** | **97.5** |

# E  Further Discussion

## E.1  Analyze Empirical Model Ensemble Rademacher Complexity

In particular, we present detailed analysis for the simple and complex cases below, within the context of transferable model ensemble attack.

**The simple input space.**  Firstly, consider the trivial case where the input space contains too simple examples so that all classifiers correctly classify $(x, y) \in \mathcal{Z}$. Then there holds $\mathcal{R}_N(\mathcal{Z}) = \ell(y, y) \underset{\sigma}{\mathbb{E}} \left[ \frac{1}{N} \sum_{i=1}^{N} \sigma_i \right] = 0$. In this case, $\mathcal{Z}$ is simple enough for $f_1, \cdots, f_N$. Such $\mathcal{Z}$ corresponds to a $\mathcal{R}_N(\mathcal{Z})$ close to 0. However, it is important to note that an

overly simplistic space $\mathcal{Z}$ may be impractical for model ensemble attack: the adversarial examples in such a space may not successfully attack the models from $D$, leading to a small value of $L_P(z^*)$. In other words, the existence of transferable adversarial examples implicitly imposes constraints on the minimum complexity of $\mathcal{Z}$.

**The complex input space.** Secondly, we consider the complex case. In particular, given arbitrarily $N$ models in $\mathcal{H}$ and any assignment of $\boldsymbol{\sigma}$, a sufficiently complex $\mathcal{Z}$ contains all kinds of examples that make $\mathcal{R}_N(\mathcal{Z})$ large: (1) If $\sigma_i = +1$, there are adversarial examples that can successfully attack $f_i$ and leads to a large $\sigma_i \ell(f_i(x), y)$; (2) If $\sigma_i = -1$, there exists some examples that can be correctly classified by $f_i$, leading to $\sigma_i \ell(f_i(x), y) = 0$. However, such a large $\mathcal{R}_N(\mathcal{Z})$ is also not appropriate for transferable model ensemble attack. It may include adversarial examples that perform well against $f_1, \cdots, f_N$ but are merely overfitted to the current $N$ surrogate models (Rice et al., 2020; Yu et al., 2022). In other words, these examples might not effectively attack other models in $\mathcal{H}$, thereby limiting their adversarial transferability. The above analysis suggests that an excessively large or small $\mathcal{R}_N(\mathcal{Z})$ is not suitable for adversarial transferability. So we are curious to investigate the correlation between $\mathcal{R}_N(\mathcal{Z})$ and adversarial transferability, which comes to the analysis about the general case in Section 3.4.

**Explain robust overfitting.** After a certain point in adversarial training, continued training significantly reduces the robust training loss of the classifier while increasing the robust test loss, a phenomenon known as robust overfitting (Rice et al., 2020; Yu et al., 2022) (also linked to robust generalization (Schmidt et al., 2018; Yin et al., 2019)). From the perspective in Section 3.4, the cause of this overfitting is the *limited complexity of the input space relative to the classifier* used to generate adversarial examples during training. The adversarial examples become too simple for the model, leading to overfitting. To mitigate this, we could consider generating more "hard" and "generalizable" adversarial examples to improve the model's generalization in adversarial training. For a less transferable adversarial example $(x, y)$, it is associated with a small $L_P(z)$, which in turn makes $TE(z, \epsilon)$ large.

## E.2 Other Opinions on "Diversity"

### E.2.1 OTHER DEFINITIONS

There are other definitions of "Diversity" in transferable model ensemble adversarial attack. For example, in Yang et al. (2021), gradient diversity is defined using the cosine similarity of gradients between different models, and instance-level transferability is introduced, along with a bound for transferability. They use Taylor expansion to establish a theoretical connection between the success probability of attacking a single sample and the gradients of the models. In Kariyappa & Qureshi (2019), inspired by the concept of adversarial subspace (Tramèr et al., 2017), diversity is defined based on the cosine similarity of gradients across different models. The authors aim to encourage models to become more diverse, thereby achieving "no overlap in the adversarial subspaces," and provide intuitive insights to readers. Both papers define gradient diversity and explain its impact.

In contrast, our definition of diversity stems from the unified theoretical framework proposed in this paper. Specifically: (1) We draw inspiration from statistical learning theory (Shalev-Shwartz et al., 2010; Bartlett & Mendelson, 2002) on generalization, defining transferability error accordingly. (2) Additionally, we are motivated by ensemble learning (Abe et al., 2023; Wood et al., 2024), where we define diversity as the variation in outputs among different ensemble models. (3) Intuitively, when different models exhibit significant differences in their outputs for the same sample, their gradient differences during training are likely substantial as well. This suggests a potential connection between our output-based definition of diversity and the gradient-based definitions in previous work, which is worth exploring in future research.

### E.2.2 CONFLICTING OPINIONS

We observe a significant and intriguing disagreement within the academic community concerning the role of "diversity" in transferable model ensemble attacks: Some studies advocate for enhancing model diversity to produce more transferable adversarial examples. For instance, Li et al. (2020) applies feature-level perturbations to an existing model to potentially create a huge set of diverse "Ghost Networks". Li et al. (2023) emphasizes the importance of diversity in surrogate models and promotes attacking a Bayesian model to achieve desirable transferability. Tang et al. (2024) supports the notion of improved diversity, suggesting the generation of adversarial examples independently from individual models. In contrast, other researchers adopt a diversity-reduction strategy to enhance adversarial transferability. For example, Xiong et al. (2022) focuses on minimizing gradient variance among ensemble models to improve transferability. Meanwhile, Chen et al. (2023) introduces a disparity-reduced filter designed to decrease gradient variances among surrogate models in ensemble attacks. Although all these studies reference "diversity," their perspectives appear to diverge. In this paper, we advocate for increasing

the diversity of surrogate models. However, we also recognize that diversity-reduction approaches have their merits. For instance, consider the vulnerability-diversity decomposition of transferability error presented in Theorem 4.1. It suggests the presence of a vulnerability-diversity trade-off in transferable model ensemble attacks. In other words, we may need to prioritize either vulnerability or diversity to effectively reduce transferability error. Diversity-reduction approaches aim to stabilize the training process, thereby increasing the "bias." In contrast, diversity-promoting methods directly enhance "diversity." This analysis, framed within our unified theoretical framework, provides insight into the differing opinions regarding adversarial transferability in the academic community.

### E.3    Compare with A Previous Bound

Lemma 5 in Yang et al. (2021) offer complementary perspectives in the analysis of transferable adversarial attack. We first restate Lemma 5 in Yang et al. (2021) and our Theorem 4.1. Our theoretical results and theirs offer complementary perspectives in the analysis of transferable adversarial attack.

**Lemma 5** (Yang et al. (2021)). *Let $f, g : \mathcal{X} \to \mathcal{Y}$ be classifiers, $\delta, \rho, \epsilon \in (0, 1)$ be constants, and $\mathcal{A}(\cdot)$ be an attack strategy. Suppose that $f, g$ have risk at most $\epsilon$. Then*

$$\Pr(\mathcal{F}(\mathcal{A}(x)) \neq \mathcal{G}(\mathcal{A}(x))) \leq 2\epsilon + \rho,$$

*for a given random instance $x$ and $\mathcal{A}(\cdot)$ is $\rho$-conservative (TV distance between the adversarial example distribution and clean data distribution is less than $\rho$, which is defined as Definition 7 and 8 in Yang et al. (2021)).*

Lemma 5 states an intriguing conclusion: if two models exhibit low risk on the original data distribution and the distributional discrepancy between adversarial examples and the original data is small, the predictions of the two models on the same input will be close. In other words, for two well-performing models, if an attack strategy successfully targets one model, it is highly likely to succeed on the other. Lemma 5 thus describes the success rate of transferring an attack from one model to another. In contrast, Theorem 4.1 demonstrates that if the ensemble models exhibit significant output differences on the same input, the resulting diverse ensemble is more effective at generating adversarial examples with reduced transferability.

To better clarify, let A denote the ensemble models generating adversarial examples and B the model being attacked. Comparing Lemma 5 and our work leads to the following reasoning: Suppose A and B both fit the original data distribution well (i.e., the risk of A and B is bounded by $\epsilon$, as in Lemma 5). As shown in our work, increasing ensemble diversity while keeping vulnerability constant reduces the transferability error of adversarial examples generated by the ensemble. Many models in parameter space, such as A and B, are vulnerable to these adversarial examples. However, fitting both the original data distribution and the adversarial example distribution simultaneously becomes challenging, leading to a large distributional discrepancy. This discrepancy enlarges $\rho$ in Lemma 5, thereby loosening its "conservative condition" and weakening its theoretical guarantee of successful transferability. Consequently, adversarial transferability decreases, which could be interpreted as a potential contradiction.

No actual contradiction exists between Lemma 5 and our work. Instead, they provide complementary analyses. Lemma 5 provides an upper bound rather than an equality or lower bound. While an increase in $\rho$ loosens this upper bound, it does not necessarily imply that the left-hand side (i.e., transferability success) will increase. The significance of an upper bound lies in the fact that a tighter right-hand side suggests the potential for a smaller left-hand side. However, a looser upper bound does not necessarily imply that the left-hand side will increase. Therefore, while increasing ensemble diversity may loosen the upper bound in Lemma 5, it does not contradict the fundamental interpretation of it. While Lemma 5 analyzes the trade-off between $\epsilon$ (model fit to the original data) and $\rho$ (distributional discrepancy), our work focuses on the trade-off between vulnerability and ensemble diversity. Together, they provide a comprehensive understanding of the factors influencing adversarial transferability.

We now further elucidate the relationship between our results and Lemma 5. To minimize transferability error (as in our work), the adversarial transferability described by Lemma 5 may have stronger theoretical guarantees, requiring its upper bound to be tighter. To tighten the bound in Lemma 5, either $\epsilon$ or $\rho$ must decrease. However, the two exhibit a trade-off:

- If $\epsilon$ decreases, A and B fit the original data distribution better. However, beyond a certain point, the adversarial examples generated by A diverge significantly from the original data distribution, increasing $\rho$.

- If $\rho$ decreases, the adversarial example distribution becomes closer to the original data distribution. However, beyond a certain point, A exhibits similar losses on both distributions, resulting in a higher $\epsilon$.

Therefore, Lemma 5 indicates the potential trade-off between $\epsilon$ and $\rho$ in adversarial transferability, while our Theorem 1 emphasizes the trade-off between vulnerability and diversity. By integrating the perspectives from both Lemma 5 and our findings, these results illuminate different facets of adversarial transferability, offering complementary theoretical insights.

### E.4    Compare with Generalization Error Bound

We note that a key distinction between transferability error and generalization error lies in the *independence assumption*. Conventional generalization error analysis relies on an assumption: each data point from the dataset is independently sampled (Zou & Liu, 2023; Hu et al., 2023). By contrast, the surrogate models $f_1, \cdots, f_N$ for ensemble attack are usually trained on the datasets with similar tasks, e.g., image classification. In this case, such models tend to correctly classify easy examples while misclassify difficult examples (Bengio et al., 2009). Consequently, such correlation indicates dependency (Lancaster, 1963), suggesting that ***we cannot assume these surrogate models behave independently for a solid theoretical analysis***. Additionally, there are alternative methods for analyzing concentration inequality in generalization error analysis that do not rely on the independence assumption (Kontorovich & Ramanan, 2008; Mohri & Rostamizadeh, 2008; Lei et al., 2019; Zhang et al., 2019). However, such data-dependent analysis is either too loose (Lampert et al., 2018) (because it includes an additional additive factor that grows with the number of samples (Esposito & Mondelli, 2024)) or requires specific independence structure of data (Zhang & Amini, 2024) that may not align well with model ensemble attacks. Therefore, we uses the latest techniques of information theory (Esposito & Mondelli, 2024) about concentration inequality regarding dependency.

### E.5    Vulnerability-diversity Trade-off Curve

The relationship between vulnerability and diversity, as discussed in Section 5, merits deeper exploration. Drawing on the parallels between the vulnerability-diversity trade-off and the bias-variance trade-off (Geman et al., 1992), we find that insights from the latter may prove valuable for understanding the former, and warrant further investigation. The classical bias-variance trade-off suggests that as model complexity increases, bias decreases while variance rises, resulting in a U-shaped test error curve. However, recent studies have revealed additional phenomena and provided deeper analysis (Neal et al., 2018; Neal, 2019; Derumigny & Schmidt-Hieber, 2023), such as the double descent (Belkin et al., 2019; Nakkiran et al., 2021). Our experiments indicate that diversity does not follow the same pattern as variance in classical bias-variance trade-off. Nonetheless, there are indications within the bias-variance trade-off literature that suggest similar behavior might occur. For instance, Yang et al. (2020) proposes that variance may exhibit a bell-shaped curve, initially increasing and then decreasing as network width grows. Additionally, Lin & Dobriban (2021) offers a meticulous understanding of variance through detailed decomposition, highlighting the influence of factors such as initialization, label noise, and training data. Recent studies have even revealed that bias and variance can exhibit a concurrent relationship in deep learning models (Chen et al., 2024c). Overall, the trend of variance in model ensemble attack remains a valuable area for future research. We may borrow insights from machine learning literature (see the above papers and the references therein) to get a better understanding of this in future work.

### E.6    Insight for Model Ensemble Defense

While our paper primarily focuses on analyzing model ensemble attacks, our theoretical findings can also provide valuable insights for model ensemble defenses: (1) From a theoretical perspective, the vulnerability-diversity decomposition introduced for model ensemble attacks can likewise be extended to model ensemble defenses. Mathematically, this results in a decomposition similar to conclusions in ensemble learning (see Proposition 3 in Wood et al. (2024) and Theorem 1 in Ortega et al. (2022)), which shows that within the adversarial perturbation region, Expected loss ≤ Empirical ensemble loss − Diversity. Thus, to improve model robustness (reduce the expected loss within the perturbation region), the core strategy involves minimizing the ensemble defender's loss or increasing diversity. However, there is also an inherent trade-off between these two objectives: when the ensemble loss is sufficiently small, the model may overfit to the adversarial region, potentially reducing diversity; conversely, when diversity is maximized, the model may underfit the adversarial region, potentially increasing the ensemble loss. Therefore, from this perspective, our work provides meaningful insights for adversarial defense that warrant further analysis. (2) From an algorithmic perspective, we can consider recently proposed diversity metrics, such as Vendi score (Friedman & Dieng, 2022) and EigenScore (Chen et al., 2024a). Following the methodology outlined in Deng & Mu (2023), diversity can be incorporated into the defense optimization objective to strike a balance between diversity and ensemble loss. By finding an appropriate trade-off between these two factors, the effectiveness of ensemble defense may be enhanced.

