# OpenReview forum: "Understanding Model Ensemble in Transferable Adversarial Attack"
_ICML.cc/2025/Conference — ICML 2025 poster_

### Official Review · Reviewer_L4RB · 2025-03-06

**Overall Recommendation:** 2

**Summary:**

The authors investigated the issue of transfer attacks based on ensembles. They provided a theoretical framework for the transferability of adversarial examples, which can be controlled by theor loss and variance among models. The authors conducted some experiments to validate their theoretical findings.

**Claims And Evidence:**

Most of the claims are clear and reasonable.

**Essential References Not Discussed:**

Some theoretical papers have been overlooked, e.g., [1].

[1] Transferability Bound Theory: Exploring Relationship between Adversarial Transferability and Flatness, NeurIPS 2024.

**Experimental Designs Or Analyses:**

The experiments are limited. The authors' experimental results do not differ significantly from those in previous empirical studies. Additionally, the authors mentioned conducting experiments in ImageNet, yet I could not find any results in ImageNet. The model architectures and attack methods used in the experiments are also limited. The authors could include visual transformers and more transfer attack methods.

**Methods And Evaluation Criteria:**

This paper does not introduce new methods or datasets.

**Other Comments Or Suggestions:**

I have no further comments.

**Other Strengths And Weaknesses:**

The authors' theory appears to offer little benefit for the future development of transfer adversarial attacks. It is also unclear how tight the suggested bound is.

**Questions For Authors:**

Please refer to the above points.

**Relation To Broader Scientific Literature:**

The authors establish a theory regarding ensemble transfer adversarial attacks.

**Theoretical Claims:**

There are some flaws in the authors' theory. Firstly, in transfer attacks, the surrogate model and the target model have different model architectures and parameter numbers. However, the authors implicitly assume that the surrogate and target models share the same parameter space (right column, Line 111-112), which significantly limits the generalizability of their theoretical results. Secondly, I suggest revising some of the notation. It is strange that (x, y) represents an adversarial example (right column, Line 121-122). Finally, the theoretical results do not seem particularly novel, as they represent a widely accepted conclusion.

---

> ### Author Rebuttal · Authors · 2025-03-31
>
> Thank you very much for your constructive comments! We address all your questions and concerns in the following responses.
>
> >**Q1**: In transfer attacks, the surrogate model and the target model have different model architectures and parameter numbers. However, the authors implicitly assume that the surrogate and target models share the same parameter space (right column, Line 111-112), which significantly limits the generalizability of their theoretical results.
>
> **A1**: Thank you for your comments.
> - Firstly, in Section 4.2 (Remark 2), we explicitly discuss how Theorem 4.3 can be generalized to cases where the surrogate and target models have distinct parameter distributions. This addresses part of the concern regarding parameter space assumptions.
> - Secondly, as the first theoretical work to formally link adversarial transferability with generalization theory, our primary contribution is a foundational framework rather than an exhaustive treatment of all scenarios. To our best knowledge, transferable model ensemble adversarial attacks remain less theoretically explored. The field still lacks papers that understand it theoretically. We hope our work inspires follow-up studies to fully address the problems you mention in the future.
>
> >**Q2**: I suggest revising some of the notation. It is strange that (x, y) represents an adversarial example (right column, Line 121-122).
>
> **A2**: Thank you for your constructive comments. We agree with your suggestion and will use $(x^{\text{adv}}, y)$ instead to represent an adversarial example in our final version.
>
> >**Q3**: The theoretical results do not seem particularly novel, as they represent a widely accepted conclusion.
>
> **A3**: We sincerely appreciate the reviewer's comments. Firstly, we're encouraged that multiple other reviewers (uk2i, kypp, and Ejwu) have explicitly recognized the novelty of our theoretical framework in their comments. It suggests our approach does offer fresh insights to the field. Secondly, we theoretically analyzes and validates three important practical guidelines to improve adversarial transferability, including (1) incorporating more surrogate models, and (2) increasing their diversity, and (3) reducing their complexity in cases of overfitting. Our work provides the first theoretical framework to explain these empirical observations in adversarial transferability, while offering practical insights for algorithm design. We hope this foundation inspires deeper understanding and future advances in the field.
>
> >**Q4**: The experiments are limited. The authors' experimental results do not differ significantly from those in previous empirical studies. Additionally, the authors mentioned conducting experiments in ImageNet, yet I could not find any results in ImageNet. The model architectures and attack methods used in the experiments are also limited. The authors could include visual transformers and more transfer attack methods.
>
> **A4**: We appreciate the reviewer's constructive feedback, which has helped us strengthen our work. Our experimental design was carefully crafted to validate our novel theoretical contributions previously unexplored in the literature. In response to the reviewer's valuable suggestions, we have expanded our experiments to include additional attack methods, diverse model architectures (including transformers) and larger dataset (ImageNet). These new experiments, detailed in our response to Reviewer Ejwu's Question 3, provide important insights for future research in transfer adversarial attacks. Furthermore, in our response to Reviewer uk2i's Question 1, we have also expanded the experimental validation by employing another effective attack methodology. This additional analysis not only confirms our theoretical findings but also broadens the scope and strengthens the practical implications of our research.
>
> >**Q5**: Some theoretical papers have been overlooked, e.g., [1].
> >
> >[1] Transferability Bound Theory: Exploring Relationship between Adversarial Transferability and Flatness, NeurIPS 2024.
>
> **A5**: We appreciate you bringing this work to our attention. We will ensure proper citation of [1] in our final version. While [1] examines flatness and transferability, we establish the theoretical link between statistical learning theory and adversarial transferability. Our framework provides new insights into transferability through the lens of generalization theory, offering complementary (rather than overlapping) perspectives to [1]. We’re happy to further clarify these differences if needed.

---

> > ### Comment · Reviewer_L4RB · 2025-04-02
> >
> > I appreciate the authors' response.
> >
> > However, I find the theoretical contributions of the proposed variant insufficiently compelling for generalizing bounds across diverse model architectures and parameterizations. The domain distance (Line 1816) is defined as maximum per-sample loss between the proxy model and the target model over the sample space. Any adversarial difference is inherently bounded by the domain distance, making the resulting bound appear vacuous in practice (Line 1831).
> >
> > Moreover, the empirical validation is inadequate. The authors should benchmark the proposed method against SOTA attacks and conduct isolated evaluations rather than just combined evaluation ("proposed method + existing techniques").
> >
> > [1] Learning to transform dynamically for better adversarial transferability, CVPR 2024.
> >
> > [2] Transferability Bound Theory: Exploring Relationship between Adversarial Transferability and Flatness, NeurIPS 2024.

---

> > > ### Author Response · Authors · 2025-04-06
> > >
> > > Thank you for your feedback on our theoretical analysis in Appendix D.4.2. We emphasize that Appendix D.4.2 is not intended as a core contribution of our work, and it can be deleted without affecting the overall contribution of this paper because Appendix D.4.1 serves a similar role as Appendix D.4.2.
> > >
> > > Regarding the experiments, our method consistently enhances SVRE across diverse model architectures, and SVRE itself is one of the SOTA attack methods in adversarial transferability. The methods in [1-2] are also not ensemble attack algorithms and cannot be compared in our validation experiments. While our focus of this paper is theory, the experiments regarding an effective attack algorithm that outperforms SOTA is out of scope of this work.
> > >
> > > We will also ensure that [2] is properly cited in Section 2.1, 2.2 and Appendix C.1 in our revision. We will clarify that [2] represents the first theoretical study in this field (although our theoretical framework remains fundamentally distinct from theirs).
> > >
> > > [1] Learning to transform dynamically for better adversarial transferability, CVPR 2024.
> > >
> > > [2] Transferability Bound Theory: Exploring Relationship between Adversarial Transferability and Flatness, NeurIPS 2024.
> > >
> > > **____________________________________________________________________________________________________________________________________________________**
> > >
> > > Finally, we reiterate the key contributions of our work, and we genuinely hope to earn your support.
> > >
> > > The primary objective of our research is to construct a theoretical framework that bridges statistical learning theory with adversarial transferability. As demonstrated in Section 4.3, the analogy between them have already inspired numerous studies to develop innovative attack algorithms. Our main contributions include:
> > > 1. Introducing transferability error, diversity, and ensemble complexity as novel analytical tools for adversarial transferability research, drawing inspiration from learning theory literature [1] (Section 3 and Appendix B.1)
> > > 2. Proposing vulnerability-diversity decomposition for both squared loss and KL divergence loss, extending concepts from bias-variance decomposition [2] and ensemble learning [3] to explain the effect of ensemble attack algorithms (Section 4.1 and Appendix B.2-B.3)
> > > 3. Deriving an upper bound for ensemble complexity in adversarial transferability through analysis inspired by Rademacher complexity bounds [4] (Section 4.2 and Appendix A.1-A.4)
> > > 4. Establishing a transferability error bound using ensemble complexity and novel information-theoretic tools to address the "independent surrogate models assumption", building upon uniform convergence theory [1] and recent information theory advances [5] (Section 4.2 and Appendix B.4)
> > > 5. Developing information-theoretic analysis of transferability error inspired by information-theoretic generalization error analysis [6] (Section 4.2 and Appendix B.5)
> > >
> > > These novel contributions systematically extend and unify results from those papers spanning 1992 to 2024 within a cohesive theoretical framework. We also discuss dozens of papers in adversarial transferability in recent years.
> > >
> > > Beyond these theoretical innovations, we have:
> > > 1. Conducted comprehensive validation experiments across three datasets to substantiate our theoretical claims, supplemented by additional experiments including the evaluation for another attack algorithm, the role of disjoint training set, a further explanation of ensemble model complexity and a practical algorithm demonstration (Section 5.1-5.2 and rebuttal)
> > > 2. Provided intuitive examples, extensive analyses, and thorough discussions of related works to elucidate the connections between our findings and existing understanding of adversarial transferability (Appendix D.1-D.8)
> > > 3. Extended our theoretical framework through preliminary explorations of alternative parameter spaces (Appendix D.4.1-D.4.2)
> > >
> > > Given the novel theoretical contributions of our work, the additional experimental validation provided during rebuttal, and the consistent positive evaluation of Reviewer uk2i, kypp and Ejwu, we sincerely hope you recognize the significance of this paper.
> > >
> > > [1] Rademacher and gaussian complexities: Risk bounds and structural results. JMLR 2002.
> > >
> > > [2] Neural networks and the bias/variance dilemma. Neural computation 1992.
> > >
> > > [3] Diversity and generalization in neural network ensembles. AISTATS 2022.
> > >
> > > [4] Size-independent sample complexity of neural networks. COLT 2018.
> > >
> > > [5] Concentration without independence via information measures. TIT 2024.
> > >
> > > [6] Information-theoretic analysis of generalization capability of learning algorithms. NeurIPS 2017.

---

### Official Review · Reviewer_Ejwu · 2025-03-07

**Overall Recommendation:** 3

**Summary:**

The paper presents a theoretical framework for model ensemble adversarial attacks, focusing on transferable adversarial examples. It defines transferability error, diversity, and Rademacher complexity, and decomposes transferability error into vulnerability and diversity. The authors apply information theory to derive bounds on transferability error and suggest practical strategies for improving adversarial transferability. The framework is validated through experiments on multiple datasets.

**Claims And Evidence:**

Yes.

**Essential References Not Discussed:**

No.

**Experimental Designs Or Analyses:**

Yes.

**Methods And Evaluation Criteria:**

Yes.

**Other Comments Or Suggestions:**

Suggestions：

1. The authors should include direct comparisons with state-of-the-art adversarial attack methods in the experiments, especially focusing on black-box attacks and model ensemble techniques, to highlight the advantages of the proposed approach.
2. The authors should consider simplifying the mathematical derivations and providing more intuitive explanations or examples, making the theoretical framework more accessible to a broader audience.
3. The authors should provide more experimental validation of the method’s performance in real-world adversarial tasks, and compare it directly with other methods to establish its practical value.
4. To improve the paper's usability, the authors should consider releasing the code and providing practical examples to help other researchers implement and test the framework.

**Other Strengths And Weaknesses:**

Strengths：

1. The paper introduces a new theoretical framework to understand the role of model ensembles in transferable adversarial attacks. This framework combines Rademacher complexity and information-theoretic tools, which adds some theoretical novelty.
2. The authors provide a mathematical decomposition of the transferability error in model ensemble adversarial attacks, highlighting the trade-offs between vulnerability and diversity. This new perspective contributes to the understanding of adversarial transferability.
3. The experimental results are comprehensive and validate the theoretical claims across multiple datasets and model architectures.

Weaknesses：

1. While the framework is theoretically innovative, its practical effectiveness remains unclear. The paper does not provide a sufficient comparison with existing state-of-the-art adversarial attack methods, nor does it demonstrate the practical performance of the method.
2. The paper could benefit from a more detailed discussion of the practical implications of the theoretical results, particularly in terms of designing more effective adversarial attack algorithms.
3. The experiments could be expanded to include a wider range of model architectures and datasets to further strengthen the empirical validation.
4. The mathematical derivations in the paper are highly complex and lack intuitive explanations, which may pose difficulties for readers, especially those without a deep background in information theory. To improve its practical applicability, the paper should provide a clearer connection between the theoretical framework and real-world use cases.
5. Although the paper presents a new theoretical framework, it does not sufficiently compare it to current adversarial attack methods, such as gradient-based attacks, input transformation techniques, or other model ensemble methods. Without demonstrating that the proposed framework performs better than existing approaches, its contribution remains unclear.
6. Despite the theoretical contributions, the paper fails to demonstrate significant practical benefits or applications. The framework's real-world impact is not well established.

**Questions For Authors:**

See weaknesses and suggestions above.

**Relation To Broader Scientific Literature:**

The key contributions of the paper are well-related to the broader scientific literature. The authors build on existing work in adversarial attacks and model ensemble methods, providing a theoretical foundation for transferable model ensemble attacks. They reference relevant studies on adversarial transferability, model ensemble diversity, and generalization in machine learning. The paper also discusses the connection between adversarial transferability and model generalization, drawing insights from statistical learning theory.

**Theoretical Claims:**

Yes.

---

> ### Author Rebuttal · Authors · 2025-03-31
>
> Thank you very much for your insightful review of our work!
>
> >**Q1**: The authors should include direct comparisons with state-of-the-art adversarial attack methods in the experiments.
>
> **A1**: We sincerely appreciate the reviewer’s constructive feedback. In direct response to Reviewer uk2i’s Question 1, we have significantly expanded our evaluation to incorporate additional attack algorithms as suggested.
>
> >**Q2**: The mathematical derivations in the paper are highly complex and lack intuitive explanations, which may pose difficulties for readers, especially those without a deep background in information theory.
>
> **A2**: Thank you for your constructive comments. We will provide more step-by-step explanations about information-theoretic analysis in Appendix B.5 in the final version.
>
> >**Q3**: The paper could benefit from a more detailed discussion of the practical implications of the theoretical results, particularly in terms of designing more effective adversarial attack algorithms.
>
> **A3**:  We thank the reviewer for the constructive suggestion. We show an example here to investigate how properly controlling the model complexity of surrogate models can contribute to more effective adversarial attack algorithms, which is in line with our theory. We use ImageNet as the dataset here. To conduct model ensemble attack, we fine-tune surrogate models—VGG16, InceptionV3, and Visformer—using a sparse Softmax cross-entropy loss [1]. This modification encourages sparsity in the model’s output distribution, and we observe in our experiments that the model complexity (L2 norm of the weight matrix) are reduced after using such a loss:
>
> |                     | VGG16 | Visformer | InceptionV3 |
> |:-------------------:|:-----:|-----------|:-----------:|
> |       Original      | 37.37 | 25.94     |    49.24    |
> | Sparse Softmax Loss | 33.12 | 20.6      |    48.53    |
>
> Building upon three advanced transfer attack methods—MI-FGSM [2], SVRE [3], and SIA [4]—we propose their sparsity-enhanced variants (MI-FGSM-S, SVRE-S, and SIA-S) through the integration of sparse Softmax loss during surrogate model training. We consider eight kinds of model architectures and measure attack performance via attack success rate, where higher value corresponds to stronger attack effectiveness.
>
> |   | ResNet50 | VGG16  | MobileNetV2 | InceptionV3 | ViT-B16 | PiT-B  | Visformer | Swin-T |
> |-----------|----------|--------|-------------|-------------|---------|--------|-----------|--------|
> | MI-FGSM   | 66.0     | **99.9**   | 76.8        | 97.5        | 37.3    | 53.8   | 88.9      | 66.7   |
> | **MI-FGSM-S** | **68.9**     | 99.7   | **79.2**        | **99.1**        | **39.0**    | **54.5**   | **90.6**      | **68.1**   |
> | SVRE      | 65.2     | **99.9**   | 79.0        | 98.6        | 32.4    | 49.2   | 92.3      | 64.3   |
> | **SVRE-S**    | **66.9**     | **99.9**   | **81.2**        | **98.9**        | **34.2**    | **51.3**   | **93.0**      | **65.9**   |
> | SIA       | 97.2     | **100.0**  | **98.4**        | **99.7**        | 75.9    | 91.9   | 99.0      | 96.1   |
> | **SIA-S**     | **98.1**     | **100.0**  | 98.2        | 99.6        | **79.2**    | **93.2**   | **99.5**      | **97.5**   |
>
> As can be seen in the table, these variants outperform their standard counterparts in most cases, demonstrating the benefit of controlling model complexity in both CNNs and visual transformer settings to improve adversarial transferability. Beyond this example shown above, we believe that our work can also inspire the development of more stronger attack algorithms in the future.
>
> [1] From Softmax to Sparsemax: A Sparse Model of Attention and Multi-Label Classification. ICML 2016
>
> [2] Boosting Adversarial Attacks with Momentum. CVPR 2018.
>
> [3] Stochastic Variance Reduced Ensemble Adversarial Attack
> for Boosting the Adversarial Transferability. CVPR 2022.
>
> [4] Structure Invariant Transformation for better Adversarial Transferability. ICCV 2023.
>
> >**Q4**: To improve the paper's usability, the authors should consider releasing the code and providing practical examples to help other researchers implement and test the framework.
>
> **A4**: We appreciate the reviewer's valuable suggestion. We are happy to release the code and provide implementation examples to facilitate reproducibility and adoption by the research community.

---

> > ### Comment · Reviewer_Ejwu · 2025-04-04
> >
> > Thank you for your detailed response and clarifications. I appreciate the effort to address my concerns, particularly the expanded experiments and additional explanations. While the theoretical contributions are valuable, I still feel the experimental section could be more comprehensive. As shown in the table, the improvements under the three advanced transfer attack methods and their sparsity-enhanced variants remain limited. Given the theoretical focus of the paper and the modest experimental gains, I am inclined to maintain my current evaluation. The work has merit, but it would benefit from more substantial and comprehensive experimental validation in future iterations.

---

> > > ### Author Response · Authors · 2025-04-06
> > >
> > > We sincerely appreciate the reviewer's constructive feedback and continued positive evaluation of our work. As the reviewer rightly noted, our paper primarily focuses on theoretical contributions. The novelty and significance of our theoretical framework has also been acknowledged by other reviewers like uk2i and kypp. Beyond our theoretical innovations, we have also:
> > > 1. (Section 5.1-5.2) Conducted comprehensive validation experiments across three datasets to substantiate our theoretical claims.
> > > 2. (Rebuttal) Provided additional experiments, including
> > >     - the evaluation for another attack algorithm,
> > >     - the role of disjoint training set,
> > >     - a further explanation of ensemble model complexity,
> > >     - a practical algorithm demonstration (our method consistently enhances SVRE across diverse model architectures. Notably, SVRE itself is one of the SOTA attack methods in adversarial transferability).
> > > 3. (Appendix D.1-D.8) Provided intuitive examples, extensive analyses, and thorough discussions of related works to elucidate the connections between our findings and existing understanding of adversarial transferability.
> > > 4. (Appendix D.4.1-D.4.2) Extended our theoretical framework through preliminary explorations of alternative parameter spaces.
> > >
> > > We sincerely appreciate your valuable feedback and will revise our paper according to your suggestions, such as incorporating the additional experiments from the rebuttal. Thank you once again for your time and insightful comments.

---

### Official Review · Reviewer_kypp · 2025-03-14

**Overall Recommendation:** 3

**Summary:**

The paper provides a theoretical study on transferability of model ensemble adversarial attacks. The authors formulate the problem by considering the expected value of the attacked loss over the distribution of model ensemble (equation 1) and the averaged attacked loss over the set of considered models (equation 2). The goal is to use concentration and uniform convergence analysis to bound the transferability error in Definition 3.1 through bounding the gap between (1) and (2) over all input samples (Lemma 3.2). The authors define the model ensemble Rademacher complexity in (8) and bound it in Lemma 4.2. In Theorem 4.3, they connect the bound on the defined Rademacher complexity to bound the transferability error. Section 5 includes experimental results supporting the theorems.

**Claims And Evidence:**

Mostly yes. First, let me say that I like the authors' idea on how to extend the mathematics of uniform convergence analysis in statistical learning theory to the transferability on model ensemble attacks. The theorems look correct and make sense to me.

The only gap that I can find in the authors' analysis is the term $H_{\alpha}^{\frac{1}{\alpha}}(P_{\theta^N}\Vert P_{\otimes_{i=1}^n \theta})$. The main difference between uniform convergence analysis in statistical learning theory and the authors' formulation is that in generalization analysis, we commonly assume the samples are drawn independently from a distribution, and so there is no term $H_{\alpha}^{\frac{1}{\alpha}}(P_{\theta^N}\Vert P_{\otimes_{i=1}^n \theta})$. However, in the standard model ensemble attack scenario, the models may have been trained with fully or partially identical training data, and therefore the models could be quite correlated. Therefore, it seems to me that the term $H_{\alpha}^{\frac{1}{\alpha}}(P_{\theta^N}\Vert P_{\otimes_{i=1}^n \theta})$ could be quite large and make the bound vaccuous in practice.

I suggest the authors to clearly discuss the above point in the paper, because I find it a major difference between the two problem settings. However, I still tend to rate the work positively, as I find the authors' idea very interesting.

**Essential References Not Discussed:**

No missing key reference as far as I can tell

**Experimental Designs Or Analyses:**

One key aspect of the experimental design that needs discussion is the correlation between ensemble models due to shared training data. Have the authors examined how the results change if disjoint training sets are used for the ensemble? Based on the theoretical framework, using disjoint training sets should reduce the correlation term, thereby improving transferability. I encourage the authors to conduct and report such an experiment.

**Methods And Evaluation Criteria:**

The experimental methodology appears well-structured and aligned with the theoretical claims.

**Other Comments Or Suggestions:**

See my previous comments.

**Other Strengths And Weaknesses:**

See my previous comments.

**Questions For Authors:**

See my previous comments.

**Relation To Broader Scientific Literature:**

The work extends the mathematics of uniform convergence analysis to the setting of adversarial transferability in model ensembles. The results parallel standard generalization bounds and adapt them to this new context.

**Theoretical Claims:**

While I have not verified every derivation in full detail, the results appear correct and consistent with existing theoretical techniques.

---

> ### Author Rebuttal · Authors · 2025-03-31
>
> Thank you very much for your insightful review of our work!
>
> >**Q1**: ...However, in the standard model ensemble attack scenario, the models may have been trained with fully or partially identical training data, and therefore the models could be quite correlated. Therefore, it seems to me that the term $H\_\alpha^{\frac{1}{\alpha}}\left(\mathcal{P}\_{\Theta^N} \\| \mathcal{P}\_{\bigotimes\_{i=1}^N \Theta} \right)$ could be quite large and make the bound vaccuous in practice.
>
> **A1**: To make it clear and easy to understand, we provide an intuitive approximation below. We choose $\alpha=10$, $\delta=0.01$, $\beta=1$ in our Theorem 4.3. Let $P=\mathcal{P}\_{\Theta^N}$ and $Q=\mathcal{P}\_{\bigotimes\_{i=1}^N \Theta}$. We consider the model parameters for a given precision so that $P$ and $Q$ are discrete distributions.
> - Equation (8) from [1] tells us that $H\_\alpha(P \\| Q)=e^{(\alpha-1)D\_\alpha(P,Q)}$, where $D\_\alpha(P,Q)$ is the Rényi divergence.
> - Let $\delta \in [0,1]$ be the TV distance between $Q$ and $P$, and $\beta_1=\min\_{a \in \mathcal{A}} \frac{Q(a)}{P(a)}$ be defined in Equation (8) from [2], i.e., the minimum of the ratio of the probability density function of distributions $Q$ and $P$.
> - Now we approximate $\beta\_1$. Consider there are $t$ parameter configurations for each model. For simplicity, we assume that part of the models ($f(N)$ models) play a key role in adversarial transferability, and the other $N-f(N)$ models are random sampled from these $f(N)$ models.
>     - For the product of marginal distribution $Q$, the parameters from each model are random. Consider the case of uniform distribution, where every parameter in the $N$ models share the same probability, i.e., $Q(a)=\frac{1}{t^N}$.
>     - For the joint distribution $P$, we also consider the case of uniform distribution, where $f(N)$ models are fixed and $N-f(N)$ models are randomly sampled, i.e., $P(a)=\frac{1}{t^{N-f(N)}}$.
>     - Therefore, $\beta\_1 \approx\frac{Q(a)}{P(a)}=t^{-f(N)}$, which is less than 1.
> - Substitute the above into Theorem 3 from [2], we have
> $$H\_\alpha(P \\| Q) \le 1+\frac{\delta\left(\beta\_1^{-1}-1\right)}{1-\beta\_1} \le \beta\_1^{-1} \approx t^{f(N)}$$
> - Substitute the above into Theorem 4.3 in our paper, we have
> $$\sqrt{\frac{18 \gamma \beta^2}{N} \ln \frac{2^{2+\frac{1}{\gamma}} H\_\alpha^{\frac{1}{\alpha}}\left(\mathcal{P}\_{\Theta^N} \\| \mathcal{P}\_{\Theta\_{i=1}^N \Theta}\right)}{\delta}} \le \sqrt{\frac{20}{N} \ln \left(800 \cdot t^{\frac{f(N)}{10}}\right)} \approx \sqrt{\frac{140}{N} + \frac{2
> f(N)}{N} \ln t}$$
>
> Here are several cases:
> 1. $f(N)=\mathcal{O}(N^{s})$, where $s \in (0,1)$
> 2. $f(N)=\mathcal{O}(\ln N)$
> 3. $f(N)=s N$, where $s \in (0,1)$
>
> For Cases 1 and 2, the above term asymptotically converges to zero as N becomes large. Notably, the true Hellinger term may be smaller than our derived upper bound above. Quantifying the core subset of models $f(N)$ that dominate ensemble attack performance presents a theoretically profound and practically significant research direction. This problem is particularly well-suited for future exploration, as it could fundamentally advance our understanding of transferable adversarial model ensemble attacks.
>
> [1] https://arxiv.org/pdf/2303.07245. TIT 2024.
>
> [2] https://arxiv.org/pdf/1503.03417. arXiv preprint.
>
> >**Q2**: ...Based on the theoretical framework, using disjoint training sets should reduce the correlation term, thereby improving transferability. I encourage the authors to conduct and report such an experiment.
>
> **A2**: Thank you for your valuable insight!
>
> As suggested by the reviewer, we evaluate three settings:
> - Full: All models are trained on the full training set.
> - Split: Models are trained on disjoint partitions of the training data.
> - Split-FT: Models are first trained on disjoint data partitions and then fine-tuned on the full dataset.
>
> We conduct experiments on MLP across two datasets. We measure attack performance via accuracy, where lower accuracy corresponds to stronger attack effectiveness.
>
> |            | MNIST |        |        | Fashion MNIST |        |        |
> |:----------:|:-----:|--------|--------|---------------|--------|--------|
> | $\epsilon$ | 8/255 | 16/255 | 32/255 |     8/255     | 16/255 | 32/255 |
> |    Split   |   **80.44**   | **55.12**      | **12.04**      | 61.59      | 36.33      | **11.83**      |
> |  Split-FT  |   81.99   | 59.31      | 15.15      | **61.57**      | **36.21**      | 12.27      |
> |    Full    |   84.26	   | 68.05      | 23.65      | 64.35      | 41.49      | 15.65      |
>
> The results in the table demonstrate that employing disjoint training sets (Split) indeed enhances adversarial transferability and reducing model accuracy, consistent with our theory. Also, if we consider larger model architectures, the limited training data in each split subset may result in model underfitting, which may adversely impact attack effectiveness. We will incorporate the full results in our final version.

---

### Official Review · Reviewer_uk2i · 2025-03-15

**Overall Recommendation:** 3

**Summary:**

This paper proposes novel definitions for theoretically analyzing the adversarial transferability of adversarial attacks with a model ensemble; then, it provides three practical guidelines to improve the transferability of the model ensemble attacks. Specifically, the paper first defines the transferability error, the gap between the adversarial risks between the most transferable example $z^*$ and the adversarial example $z$ that the model ensemble outputs. The paper analyzes this transferability error in two different ways.

In the first analysis, the paper decomposes the population risk on $z$ into the vulnerability term (which measures the attack power of the ensemble attack) and the diversity term (which measures the diversity of the ensemble attack). This suggests two guidelines: improving the ensemble attack’s power and diversity are both beneficial to the transferability of the model ensemble attack.

In the second analysis, the paper provides an upper bound of transferability. This upper bound contains two terms. The first term represents the complexity of the surrogate models, and the second term decreases as the number of surrogate models increases. This upper bound gives us another guideline: having more surrogate models with less model complexity is beneficial to the transferability of the model ensemble attack.

With a set of experiments, the paper experimentally supports the theoretical findings.

**Claims And Evidence:**

The paper supports all of its claims well with proof and experimental results.

**Essential References Not Discussed:**

The paper cited the needed references well.

**Experimental Designs Or Analyses:**

The experimental designs are overall well-designed, and the analyses make sense. In particular, the analysis of the different behaviors on the CIFAR dataset seems interesting.

**Methods And Evaluation Criteria:**

I checked the evaluation methods and criteria. Although the evaluation can be improved further, the experiments are well-designed to support the theoretical findings.

**Other Comments Or Suggestions:**

1. Please clarify the effect of increasing $\lambda$ in the paper and explain how it relates to the model complexity.
2. If there are other factors that can quantitatively control model complexity, consider adding more experiments involving them.
3. Please consider experiments with other attack methods.
4. A comparison between CIFAR-10 and CIFAR-100 could also be interesting. They contain the same set of images, only more labels in CIFAR-100, but this changed the variance behaviors. Does variance on CIFAR-100 experiments decrease for lower $\lambda$?

**Other Strengths And Weaknesses:**

### Strengths
1. To the best of my knowledge, all the proposed concepts (transferability error, prediction variance, and model ensemble Rademacher complexity) are novel.
2. The decomposition of the transferability error into two interpretable terms seems impressive.
3. The paper presents all of the theoretical analysis, practical discussions, and experimental validation of the proposed idea.
4. The experiments support the findings well.

### Weaknesses
1. Only one attack method is used for the evaluation. Comparing another attack method (either extremely strong or weak) would give us more insights about the vulnerability-variance tradeoffs.
2. I’m not very convinced whether or not the experiments are enough to explore the model complexity. Only three values of $\lambda$ are used, and no other quantity controls the model complexity in the experiments.
3. The effect of increasing the $\lambda$ parameter is not clearly explained, and it is unclear whether it definitely lowers the model complexity or not. In my opinion, the empirical model ensemble Rademacher complexity is computable for low values of $N$, so the model complexity can be quantified during the experiments.

**Questions For Authors:**

* Could you explain what factors would increase/decrease the model complexity in practice?

**Relation To Broader Scientific Literature:**

This paper makes many contributions to adversarial machine learning. This paper provides theoretical understandings of model ensemble attacks and practical guidelines for ML practitioners.

**Theoretical Claims:**

I did not have enough time to check the proofs in the appendices.

---

> ### Author Rebuttal · Authors · 2025-04-01
>
> Thank you very much for your insightful review of our work!
>
> >**Q1**: ...Comparing another attack method (either extremely strong or weak) would give us more insights about the vulnerability-variance tradeoffs.
>
> **A1**: We sincerely appreciate the reviewer's constructive suggestion. We have conducted additional experiments using the VMI-FGSM attack [1] on MNIST.
>
> $\lambda=10^{-4}$
>
> | Steps | 1     | 3     | 6     | 9     | 12    | 16    | 19    | 22    | 25    |
> |-------|-------|-------|-------|-------|-------|-------|-------|-------|-------|
> | ASR   | 2.1   | 6.9   | 8.4   | 17.1  | 24.6  | 29.5  | 37.3  | 38.4  | 40.2  |
> | loss  | 0.012 | 0.044 | 0.225 | 0.351 | 0.378 | 0.390 | 0.392 | 0.397 | 0.401 |
> | Variance | 0.007 | 0.025 | 0.033 | 0.019 | 0.008 | 0.006 | 0.004 | 0.003 | 0.003 |
>
> $\lambda=10^{-3}$
>
> | Steps    | 1     | 3     | 6     | 9     | 12    | 16    | 19    | 22    | 25    |
> |-----------|-------|-------|-------|-------|-------|-------|-------|-------|-------|
> | ASR  | 2.2   | 7.5   | 8.3   | 16.9  | 25.4  | 30.1  | 38.2  | 38.9  | 40.6  |
> | loss | 0.011 | 0.039 | 0.214 | 0.337 | 0.365 | 0.381 | 0.385 | 0.392 | 0.399 |
> | Variance | 0.004 | 0.017 | 0.029 | 0.015 | 0.006 | 0.005 | 0.003 | 0.003 | 0.003 |
>
> The observed vulnerability-variance tradeoffs demonstrate consistent alignment with our paper. We will include the complete experimental details and analysis on other datasets in our final manuscript.
>
> [1] Enhancing the transferability of adversarial attacks through variance tuning. CVPR 2021.
>
> >**Q2**: Please clarify the effect of increasing $\lambda$ in the paper and explain how it relates to the model complexity...
>
> **A2**: Thank you for your insightful question. Firstly, as suggested in Lemma 4.2 and Appendix A, the empirical model ensemble Rademacher complexity can be upper bounded by model complexity and the number of ensemble components (for instance, the analysis in Section 4.2 states that reducing the weight norm of the model and increasing the number of the models will reduce the empirical model ensemble Rademacher complexity). These two kinds of effect have been reported in our experiments:
> - In Section 5.1, we adjust the weight decay factor $\lambda$ to change the model complexity and investigate the trends in how complexity interacts with other factors.
> - In Section 5.2, we improve the number of ensemble components and observe an increasing trend of attack success rate.
>
> Following the reviewer’s suggestion, we conduct a deeper investigation into the impact of model complexity by applying a max norm constraint to the model parameters. This technique limits the L2 norm of each weight vector to a predefined threshold to control the model complexity.
> - Larger max norms enable richer feature representations at the cost of potential overfitting.
> - Smaller max norms promote simpler models and may induce underfitting by limiting model capacity.
>
> As illustrated in the table below, this trade-off manifests across architectures (MLPs and CNNs with 1–3 layers) and varying max norm values. We measure attack performance via accuracy, where lower accuracy corresponds to stronger attack effectiveness.
>
> | Max norm | FC1    | FC2    | FC3    | CNN1  | CNN2  | CNN3  | Avg   |
> |----------|--------|--------|--------|-------|-------|-------|-------|
> | 0.1      | 84.66  | 87.80  | 85.39  | 97.57 | 98.31 | 98.59 | 92.05 |
> | 0.5      | 59.37  | 68.31  | 74.05  | 96.50 | 97.66 | 98.34 | 82.37 |
> | 1.0      | 64.31  | 55.27  | 57.12  | 95.37 | 97.08 | 97.93 | 77.85 |
> | 2.0      | 68.00  | 57.40  | 57.86  | 95.41 | 97.04 | 97.87 | 78.93 |
> | 4.0      | 68.19  | 57.94  | 58.12  | 95.53 | 97.00 | 97.85 | 79.11 |
> | 5.0      | 69.68  | 59.40  | 59.26  | 97.48 | 98.02 | 98.87 | 80.45 |
>
> The results demonstrate a clear trend: as the max norm constraint is relaxed from very small values (e.g., 0.1) to moderate levels (e.g., 5.0), the attack effectiveness first increases and then decreases. This pattern indicates that excessively restrictive constraints can impair model expressiveness, whereas an optimally tuned max norm effectively balances model complexity and representational capacity. More importantly, these findings support our paper's claim regarding the influence of the weight decay factor $\lambda$ on model complexity.
>
> >**Q3**: A comparison between CIFAR-10 and CIFAR-100 could also be interesting...
>
> **A3**: We sincerely appreciate your insightful question. As shown in our paper, the attack success rate and loss exhibit an initial increase followed by stabilization with growing steps, while the variance first rises and then declines. We fully agree with the reviewer's insightful comment regarding dataset-dependent characteristics; indeed, the inflection points vary across datasets due to their distinct characteristics. Due to the space limitations of this rebuttal, we will extend the step range and include a comparative analysis of CIFAR-10 and CIFAR-100 in Appendix E in our final version.

---

### Decision · Program_Chairs · 2025-05-01

**Decision:**

Accept (poster)

**Comment:**

Three of the four reviewers have given positive scores with detailed reviews. The authors have put a lot of effort into addressing the reviewers' questions. In particular, the only reviewer who gave a "2: Weak Reject" said during the discussion phase that "the major concern is the applicability of the proposed theory to transfer-based attacks where the surrogate and the target models differ in architecture, and the authors acknowledge that the theory cannot be extended to this scenario for now." Considering that in the context of model ensemble-based transfer attacks, theoretical analyses are very rare, the AC thought this is not a big problem that would lead to rejection.

Suggestions from the AC and SAC for improving the paper:
(1) The theoretical analyses can be significantly shortened. Specifically, the theory is a direct application of usual uniform convergence techniques, with the setting that the learned models are the random variables and the data points define the loss functions for these random variables. Given these definitions, the analysis becomes a straightforward application of standard theory (or a more advanced version of it, coming from Esposito & Mondelli, 2024). This should be made explicit in the paper. As such, a lot of the proofs could be shortened instead of rederiving well-known parts of existing proofs with the current notation. For example, Theorem 4.3 is a direct consequence of Theorem 1 of Esposito and Mondelli (2024) and the standard way of going from tail bounds to Rademacher bounds.

(2) The paper does not discuss the dependence of the models other than the Hellinger term should be made small by having as independent models as possible. For example, a simple scenario is that if all models are trained on the same dataset then -- conditioned on the dataset -- they are independent (assuming independent initialization), so we immediately get a much stronger transfer guarantee for all models which are trained on the same training data. Exploring how the random choice of the training data affects the bound would also be an interesting addition.

(3) In addition to validating the three well-known practical guidelines for reducing transferability error, the authors should discuss what new insights the proposed theory can provide into further improving transferability.